# Mutant p53 sustains serine-glycine synthesis and essential amino acids intake promoting breast cancer growth

Camilla Tombari[1,2,10], Alessandro Zannini[1,2,10], Rebecca Bertolio ⓘ[1,2], Silvia Pedretti ⓘ[3], Matteo Audano[3], Luca Triboli[1,2], Valeria Cancila[4], Davide Vacca[4], Manuel Caputo[1,2], Sara Donzelli ⓘ[5], Ilenia Segatto[6], Simone Vodret[2], Silvano Piazza[2], Alessandra Rustighi[1,2], Fiamma Mantovani[1], Barbara Belletti ⓘ[6], Gustavo Baldassarre ⓘ[6], Giovanni Blandino ⓘ[5], Claudio Tripodo ⓘ[4,7], Silvio Bicciato ⓘ[8], Nico Mitro ⓘ[3,9] & Giannino Del Sal ⓘ[1,2,7]✉

Reprogramming of amino acid metabolism, sustained by oncogenic signaling, is crucial for cancer cell survival under nutrient limitation. Here we discovered that missense mutant p53 oncoproteins stimulate de novo serine/glycine synthesis and essential amino acids intake, promoting breast cancer growth. Mechanistically, mutant p53, unlike the wild-type counterpart, induces the expression of serine-synthesis-pathway enzymes and L-type amino acid transporter 1 (LAT1)/CD98 heavy chain heterodimer. This effect is exacerbated by amino acid shortage, representing a mutant p53-dependent metabolic adaptive response. When cells suffer amino acids scarcity, mutant p53 protein is stabilized and induces metabolic alterations and an amino acid transcriptional program that sustain cancer cell proliferation. In patient-derived tumor organoids, pharmacological targeting of either serine-synthesis-pathway and LAT1-mediated transport synergizes with amino acid shortage in blunting mutant p53-dependent growth. These findings reveal vulnerabilities potentially exploitable for tackling breast tumors bearing missense *TP53* mutations.

During tumor growth, cancer cells undergo extensive metabolic rewiring in response to environmental cues and to specific biosynthetic and energetic requirements[1,2]. Tissue-specific constraints and the presence of a rigid, poorly perfused fibrotic stroma may limit nutrient supply to the expanding tumor mass[3–6]. Reprogramming of metabolic transcriptional networks downstream to oncogenic pathways supports tumor cell adaptation to nutrient fluctuations and other harsh conditions, thus defining metabolic dependencies[1,2,7]. Within the tumor ecosystem, cancer cells display the highest avidity for amino acids (AAs) and exhibit metabolic addiction to import and/or de novo synthesis of these nutrients[8–10]. AAs constitute major precursors of all biomass components including nucleotides, proteins,

[1]Department of Life Sciences, University of Trieste, 34127 Trieste, Italy. [2]International Centre for Genetic Engineering and Biotechnology (ICGEB), Area Science Park-Padriciano, 34149 Trieste, Italy. [3]DiSFeB, Dipartimento di Scienze Farmacologiche e Biomolecolari, University of Milan, Milan, Italy. [4]Tumor Immunology Unit, Department of Health Science, Human Pathology Section, School of Medicine, University of Palermo, 90133 Palermo, Italy. [5]Translational Oncology Research Unit, IRCCS Regina Elena National Cancer Institute, Rome, Italy. [6]Unit of Molecular Oncology, Centro di Riferimento Oncologico di Aviano (CRO), IRCCS, National Cancer Institute, 33081 Aviano, Italy. [7]IFOM ETS, the AIRC Institute of Molecular Oncology, Milan, Italy. [8]Center for Genome Research, University of Modena and Reggio Emilia, 41125 Modena, Italy. [9]Department of Experimental Oncology, IEO, European Institute of Oncology IRCCS, Milan, Italy. [10]These authors contributed equally: Camilla Tombari, Alessandro Zannini. ✉e-mail: gdelsal@units.it

and lipids, and represent essential substrates for energy production, antioxidant defense and signal transduction[9,11]. AAs are key to tumor growth and survival in face of cancer cell-intrinsic and -extrinsic stress conditions, in particular nutrient scarcity[8]. Indeed, dietary or pharmacological interventions targeting AA metabolism were shown to dampen tumor growth and metastasis and to increase the efficacy of chemotherapies[5,12–17].

*TP53* is the top mutated gene in human cancers and the p53 protein acts as a master transcription factor sensing cancer-related stress stimuli and nutrient alterations, coordinating multiple tumor suppressor activities, including control of metabolism[18–20]. *TP53* mutations are mostly missense, and the resulting mutant p53 proteins (hereafter mutp53) are sensitive to tumor-inherent stress[21], being stabilized and activated downstream to mechanical cues generated by stromal stiffness[22]. Remarkably, beside losing wild-type tumor suppressor activities, mutp53 can acquire oncogenic properties that support tumor growth, metastatic propensity and therapy resistance, by reprogramming cancer cell transcriptome, proteome, and metabolome[23,24]. Mutp53 was shown to exert a wide-ranging control over tumor cell metabolism, by affecting the expression and activity of several enzymes and transporters that control glucose and mevalonate metabolism, thereby improving cancer cell fitness and reshaping the tumor ecosystem with aggressive outcomes[25–28]. In this scenario, while a role of wild-type p53 (wtp53) on regulating some aspects of AA metabolism has been demonstrated, no evidence of a pro-oncogenic impact of mutp53 on this metabolic branch has been described so far.

Here we found that in breast cancer (BC) mutp53 responds to AA scarcity and controls a metabolic transcriptional program potentiating serine-glycine-one-carbon (SGOC) metabolism and intake of essential AAs (EAAs), supporting BC growth and survival in nutrient-limiting contexts. Remarkably, this metabolic branch is also sensitive to mechanical inputs in mutp53-expressing cancers. We provide evidence in BC and patient-derived tumor organoids (PDOs) that interfering with this circuit may represent an effective and specific strategy to blunt the growth of mutp53 expressing tumors.

## Results

### Mutant p53 promotes serine and glycine synthesis and intake of essential AAs in BC cells

To investigate whether mutp53 may impact de novo AAs synthesis, we performed a metabolic tracing experiment in the BC cell line MDA-MB-231 (expressing p53R280K) upon p53 silencing, using uniformly labeled [U-$^{13}$C$_6$]-Glucose. We detected a significant reduction in the incorporation of labeled glucose-derived carbon units into serine and glycine in p53-silenced as compared to control cells, while no alterations were found for other AAs (Fig. 1a) or glycolytic and tricarboxylic acid (TCA) cycle intermediates (Suppl. Fig. 1a).

De novo Ser/Gly synthesis in MDA-MB-231 cells is relatively low[13,29,30], thus we repeated [U-$^{13}$C$_6$]-Glucose tracing upon removal of Ser/Gly from the medium (-S/G) to activate this biosynthetic pathway[31,32]. Silencing of mutp53 prevented the induction of de novo synthesis of both AAs caused by growth in -S/G conditions (Suppl. Fig. 1b).

To investigate the mechanism underlying the effect of mutp53 on Ser/Gly biosynthesis, we performed RNA-seq in MDA-MB-231 cells upon p53 silencing. Transcriptomic analysis revealed a significant down-regulation of enzymes belonging to SGOC metabolism upon mutp53 silencing, in particular phosphoserine aminotransferase 1 (*PSAT1*) (responsible for serine synthesis) and serine hydroxymethyltransferase 1 (*SHMT1*) (responsible for glycine synthesis) (Fig. 1b). We also found a significant down-regulation of several AA transporters belonging to SLC1 and SLC7 families, in particular *SLC7A5* and *SLC3A2* (coding for the LAT1/CD98hc heterodimer, a transporter of EAAs as leucine, valine, methionine, and others), and *SLC1A5* (coding for ASCT2, transporter of neutral AAs as glutamine, alanine, serine, and

others[33,34]) (Fig. 1b). This analysis was confirmed at both mRNA and protein level in several BC cell lines harboring different missense mutp53 variants (MDA-MB-231[p53R280K]; MDA-MB-468[p53R273H]; SUM-149PT[p53M237I]) (Fig. 1c, d). Notably, all serine synthesis pathway (SSP) enzymes, i.e., phosphoglycerate dehydrogenase (*PHGDH*), *PSAT1*, phosphoserine phosphatase (*PSPH*), were reduced upon mutp53 silencing in both BC cells with low (MDA-MB-231)[13,29,30] and high (MDA-MB-468; SUM-149PT)[29,30,35] serine synthesis capability (Fig. 1c). Moreover, ectopic expression of different doxycycline-inducible mutp53 variants (R175H, R273H, R249S, and R280K) in p53-null mouse BC 4T1 cells significantly induced the expression of *SSP*, *SLC7A5*, and *SLC3A2* genes (Suppl. Fig. 1c). Previous work uncovered that wtp53 regulates AA metabolism enzymes and transporters, directly or through its target p21[18]. In colon cancer cells, wtp53 was shown to redirect serine usage within one-carbon metabolism while leaving the SSP unaltered, thus supporting cancer cells proliferation, a function also retained by the specific missense mutp53 variant R248W[31,36]. To investigate whether in BC cells wtp53 might impact Ser/Gly synthesis, we performed a metabolic tracing experiment in the human cell line MCF10DCIS.COM (hereafter DCIS) upon p53 silencing. We found that, unlike mutp53, silencing of wtp53 did not perturb the biosynthesis of Ser/Gly and of any other AAs, nor of glycolytic and TCA cycle intermediates (Suppl. Fig. 1d, e). Moreover, neither silencing of wtp53 in MCF7 or DCIS cells, nor its ectopic expression in 4T1 cells had significant effects on the expression of *SSP*, *SLC7A5*, and *SLC3A2* genes (Suppl. Fig. 1f, g), suggesting that the metabolic features here described are specific of mutp53 proteins.

LAT1 (*SLC7A5*), in complex with CD98hc (*SLC3A2*), constitutes a heterodimeric antiporter mediating the influx of EAAs, including branched-chain AAs (BCAAs) and aromatic AAs, in exchange for intracellular non-essential AAs (NEAAs), mainly glutamine, which is instead imported by ASCT2[33,34,37–39]. Given that mutp53 controls the expression of all three AA transporters, we analyzed whether its silencing impacts the uptake of LAT1 substrates. Metabolic tracing analysis with [U-$^{13}$C$_6$]-Leucine, a BCAA, revealed that mutp53 knockdown reduced the uptake of leucine to the same level as the LAT1-specific inhibitor JPH203[40] (Fig. 1e).

mTORC1 is a key sensor of AA abundance and activates several downstream targets (e.g., S6RP ribosomal protein S6) leading to protein synthesis and cell growth. Conversely, when AA levels are low, mTORC1 becomes inactive, slowing protein synthesis and promoting autophagy[41,42]. Consistently with the observation that mutp53 knockdown impairs EAAs intake and AAs synthesis, we detected a significant reduction in phosphorylated S6RP and a concomitant increase of the autophagosome marker LC3-II (Fig. 1f), reflecting mTORC1-dependent switch from anabolic to catabolic pathways. We next asked whether mutp53 may directly regulate the above-identified genes encoding for SGOC enzymes and AA transporters. Inspection of a mutp53 ChIP-seq performed in BC cells, available in the public resource ReMap 2022[43], highlighted that all AA metabolism genes, whose expression was influenced by mutp53 in RNAseq, displayed mutp53 binding peaks within gene regions upstream to TSS that are characterized by active promoter histone marks (Suppl. Fig. 2a and Suppl. Table 1). Given that mutp53 often binds DNA through interactions with specific transcription factors (TFs), we examined the TFs co-occurring in the same regions as the mutp53 ChIP-seq peaks for *PSAT1*, *SLC7A5* and *SLC3A2*. We identified 10 putative TFs, including MYC/MAX, which has been reported to transcriptionally regulate several AAs-related genes and to interact with mutp53 in cancer (Suppl. Table 1)[44]. By ChIP experiments in MDA-MB-231 cells we then confirmed that mutp53 associates to *PSAT1*, *SLC7A5* and *SLC3A2* gene promoters in a MYC-dependent way (Suppl. Fig. 2b). Moreover, we observed that the upregulation of *PSAT1*, *SLC7A5* and *SLC3A2* promoted by expressing doxycycline-inducible mutp53 in DCIS cells was dampened upon silencing of MYC (Suppl. Fig. 2c). This evidence indicates that MYC is required for

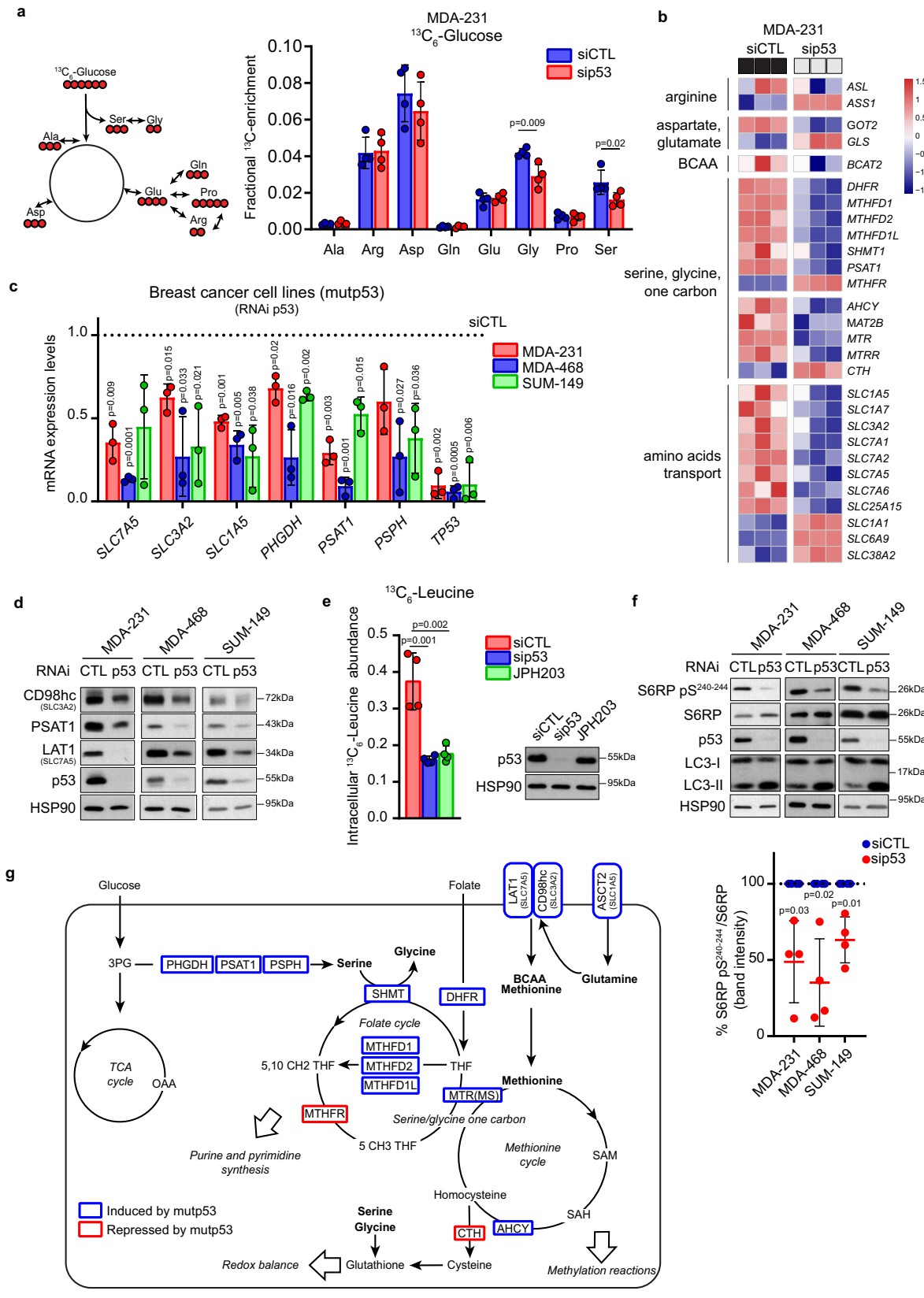

mutp53 to bind the promoter regions of *PSAT1*, *SLC7A5* and *SLC3A2* and to promote their expression.

Altogether, these results demonstrate that in BC cells missense mutp53, but not wtp53, regulates the expression of SSP enzymes promoting Ser/Gly synthesis and transporters linked to the intake of EAAs (Fig. 1g).

## Breast tumors bearing mutant p53 express high levels of SSP enzymes and LAT1/CD98hc EAAs transporter

To validate our observations in vivo, p53-null metastatic BC 4T1 cells, engineered to express doxycycline-inducible HA-tagged mutp53[R280K], were orthotopically injected into the mammary gland of syngeneic mice. As shown in Fig. 2a–c, mutp53 expression, induced by doxycycline

**Fig. 1 | Mutp53 promotes a specific AA metabolic program. a** Left panel: scheme of glucose-derived AAs where glucose-derived carbons are indicated in red. Right panel: mass isotopomer distribution (MID) of AAs from [U-$^{13}$C$_6$]-glucose in MDA-MB 231 cells upon control (siCTL) or p53 (sip53) silencing (isotopomers displayed in the scheme are those reported in the graph). For labeling experiments, cells were transfected with control or p53 siRNAs on the day of seeding and cultivated for 24 h followed by incubation with [U-$^{13}$C$_6$]-glucose in the medium for additional 24 h; $n = 4$ biological replicates. **b** Heatmap of RNA-seq data of genes related to AA biosynthesis, intake, and metabolism in MDA-MB 231 cells upon control (siCTL) or p53 (sip53) silencing. Three columns for each condition represent $n = 3$ biological replicates. **c** qRT-PCR analysis of indicated genes in MDA-MB 231, MDA-MB 468, and SUM-149 cells upon silencing of p53. mRNA levels relative to control condition (dotted line) are shown; $n = 3$ biological replicates. **d** Western blot analysis of indicated proteins in MDA-MB 231, MDA-MB 468, and SUM-149 cells upon silencing

of p53. CD98hc is encoded by *SLC3A2* and LAT1 by *SLC7A5*. HSP90 was used as loading control; $n = 3$. **e** Left panel: intracellular level of $^{13}$C$_6$-Leucine (Mass isotopomer distribution (MID) of M6 Leucine) in MDA-MB 231 cells upon control (siCTL) or p53 (sip53) silencing and treated with 10 μM JPH203 (LAT1i) as a positive control. Right panel: western blot analysis of p53 levels in the above-described condition. HSP90 was used as loading control; $n = 4$ biological replicates. **f** Upper panel: western blot analysis of indicated proteins in MDA-MB 231, MDA-MB 468, and SUM-149 cells upon silencing of p53. HSP90 was used as loading control. Lower panel: quantification of depicted western blot bands expressed as percentage of S6RP pS240-244 relative to S6RP; $n = 4$ biological replicates. **g** Schematic overview of genes encoding for AA enzymes and transporters downregulated (blue) and upregulated (red) upon silencing of mutp53. Graph bars represent mean±s.d. Two-tailed Student's *t*-test or Exact test *p*-value (FDR) cutoff 0.05. Source data are provided as a Source Data file.

administration, led to a significant increase in tumor volume along with enhanced expression of all SSP enzymes and transporters of EAAs and glutamine (*SLC7A5*, *SLC3A2*, and *SLC1A5*), and upregulation of mTORC1 signaling. Conversely, a similar experiment performed with 4T1 cells bearing doxycycline-inducible wtp53 led to reduction in tumor volume upon wtp53 induction, while no significant variation of the expression of SSP enzymes and AA transporters was detected, except for a slight increase of *SLC7A5* and *SLC3A2* mRNA levels (Suppl. Fig. 3a, b). These observations were confirmed in a panel of invasive ductal BCs ($n = 10$), stratified according to p53 protein levels and classified as either p53-high cases (considered as a marker of missense p53 mutations in the diagnostic practice) or p53-low cases[45,46]. As shown in Fig. 2d, expression of PSAT1, LAT1, CD98hc and the mTORC1 target 4EBP1 pT[37-46] was consistently higher in p53-high cases, with LAT1 being almost undetectable in p53-low cases. Furthermore, we investigated a panel of 701 primary BCs (Metabric dataset), stratified for *TP53* status (wt or missense mutations). These analyses revealed that BCs harboring missense mutp53 exhibited significantly higher expression of SSP genes (*PHGDH*, *PSAT1*, and *PSPH*), AA transporter genes (*SLC7A5*, *SLC3A2*, *SLC1A5*) and activation of mTORC1 signaling (Fig. 2e). Similar results were obtained by analyzing the TCGA BC dataset (Suppl. Fig. 3c).

To investigate the association of mutp53 with AA genes expression also in a non-tumoral setting, we employed a pre-neoplastic mouse model bearing the R172H hotspot missense mutation (equivalent to the human R175H). Cells from normal mammary glands of 8-week-old p53$^{+/+}$ and p53$^{R172H/R172H}$ mice were subjected to single-cell RNAseq. 15 distinct cell populations belonging to 3 main groups, i.e., epithelial, immune, and stromal cells, were identified (Suppl. Fig. 3d), and mutp53-expressing mammary gland displayed a reduced percentage of B cells and T cells and increased percentage of fibroblasts and basal cells (Suppl. Table 2, 3). Focusing on the epithelial population, which includes luminal cells, luminal progenitor cells, luminal differentiated cells and basal cells, functional enrichment analysis of differentially expressed genes revealed upregulation of metabolic pathways related to glucose, lipids, and AAs in cells with missense p53 mutation (Suppl. Fig. 3e). Specifically, higher expression levels of genes encoding SSP enzymes and AA transporters were observed in cells bearing mutp53 compared to those with wtp53 (Suppl. Fig. 3f and Suppl. Table 4). The observed variations however, although significant, were very small, which is not unexpected considering that, in the pre-neoplastic breast tissue, mutp53 is not stabilized and activated[47].

Altogether, these results demonstrate that missense mutp53 expression is associated with increased expression of SSP enzymes and EAAs transporters in both mouse and human BC tissues, and with a lesser extent, in preneoplastic breast tissues.

## Mutant p53 becomes stabilized and enables BC cell proliferation upon reduced AA availability

Cancer cells within a growing tumor mass often experience scarcity of oxygen and nutrients, stress conditions that prelude to oncogene-

dependent rewiring of cell metabolism to preserve proliferative potential[5,48,49]. We thus sought to investigate the behavior of tumor cells expressing oncogenic missense mutp53 in response to changes in AA availability, as compared to cells bearing wtp53. As shown in Fig. 3a, the proliferation rate of wtp53-expressing BC cells dropped significantly upon progressive reduction of AA concentration. In contrast, BC cells expressing mutp53 maintained a high proliferation rate even when cultured in presence of very low AA content (5%) for 3 days. This effect was dependent on mutp53 expression, as its silencing significantly reduced cell proliferation in low AA conditions (Fig. 3b; Suppl. Fig. 4a). In contrast, silencing of wtp53 did not significantly impact proliferation under low AA (Suppl. Fig. 4b). We compared the response of BC cells expressing either wild-type or mutp53 to AA shortage. When grown in low AA, DCIS and 4T1 cells expressing mutp53 variants exhibited increased proliferation, with 4T1 bearing p53R175H and p53R280K showing an approximately 2-fold higher replication rate as compared to those bearing empty vector or p53WT (Suppl. Fig. 4c–e).

Cells grown as 3D spheroids recapitulate in vitro the spatial and nutrient heterogeneity (e.g., oxygen and nutrients availability) occurring within a tumor mass[50]. To confirm our observations, we cultured DCIS cells expressing either endogenous wtp53 alone or doxycycline-inducible HA-tagged mutp53R280H as 3D tumor spheroids in complete medium (CM) or in low AA condition. As shown in Fig. 3c and d, DCIS spheroids expressing wtp53 exhibited a reduced replication rate (judged by BrdU labeling in Fig. 3c) and mTORC1 activation (evaluated by phosphorylation of its target 4EBP1) when grown in low AA. In contrast, replication of DCIS-mutp53R280K spheroids was less affected (Fig. 3c) and sustained activation of mTORC1 was evident (Fig. 3d, e). These results suggest that mutp53 allows cancer cells to thrive under AA starvation, by sustaining cell proliferation along with mTORC1 activation.

The oncogenic activity of mutp53 has been linked to its protein stabilization[22,51]. Interestingly, by immunofluorescence, protein levels of ectopically expressed mutp53 appeared higher in DCIS spheroids maintained in low AA than in those grown in CM (Fig. 3f), thus suggesting that AA scarcity may influence the stabilization of mutp53. Furthermore, in two different BC cell lines (MDA-MB-231 and MDA-MB-468) mutp53 protein, but not its mRNA levels, increased upon progressive AA limitation, and this was accompanied by a concomitant increase of PSAT1 levels (Fig. 3g and Suppl. Fig. 4a, f). In normal and cancer cells, AA scarcity is sensed by the Eukaryotic Translation Initiation Factor 2 Alpha Kinase 4 (eIF2AK4 also known as GCN2), which phosphorylates eukaryotic translation initiation factor 2A (eIF2α), to switch Cap-dependent/Cap-independent translation, promoting the expression of AA biosynthetic enzymes and transporters as an adaptive response[5,52]. As shown in Fig. 3h and Suppl. Fig. 4g, in low AA, we consistently observed a GCN2-dependent increased phosphorylation of eIF2α, and a reduction of mutp53 protein levels upon GCN2 inhibition. These data suggest that activation of AA stress response

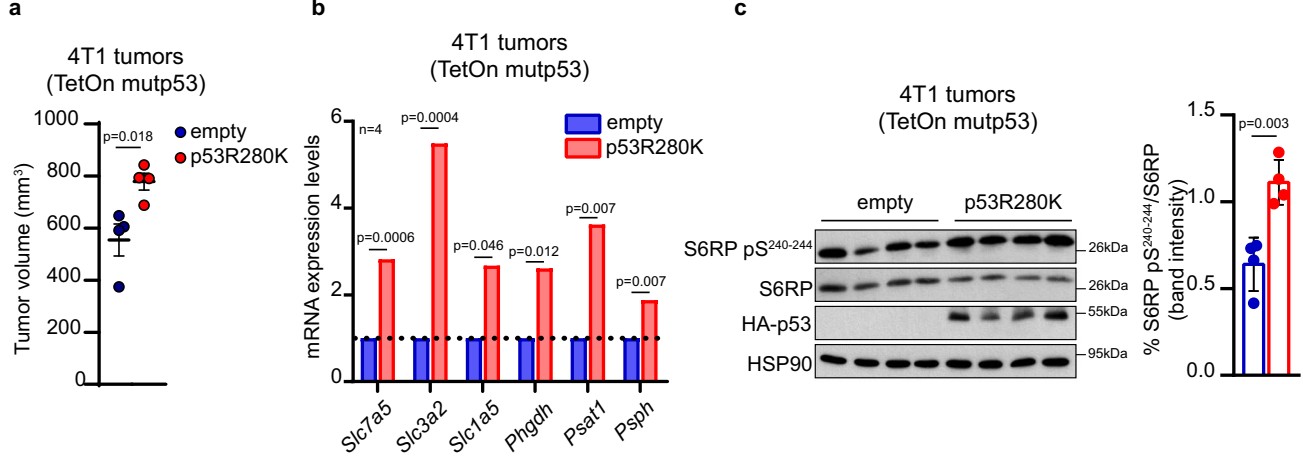

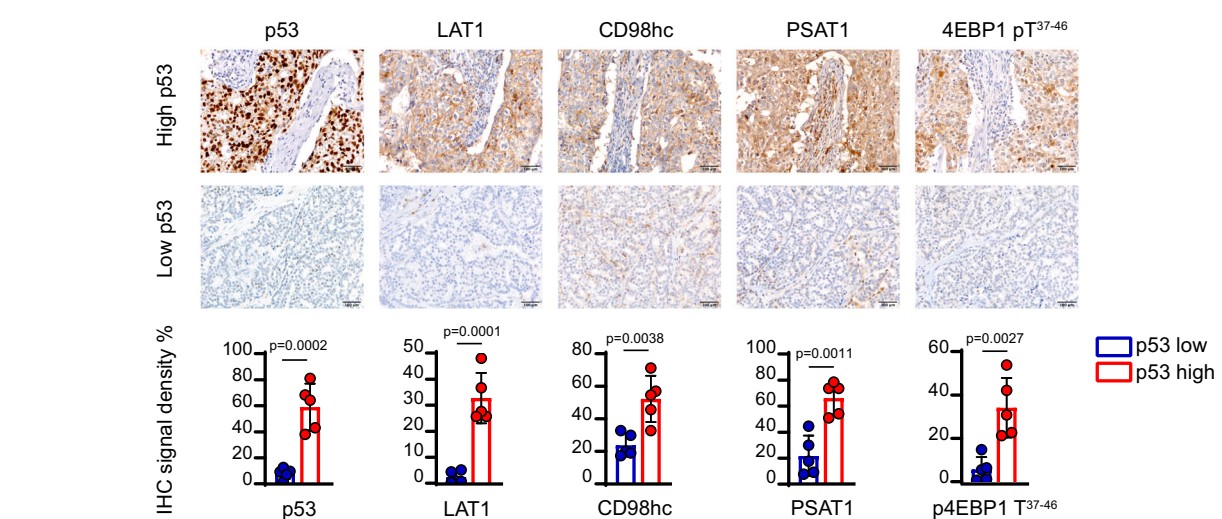

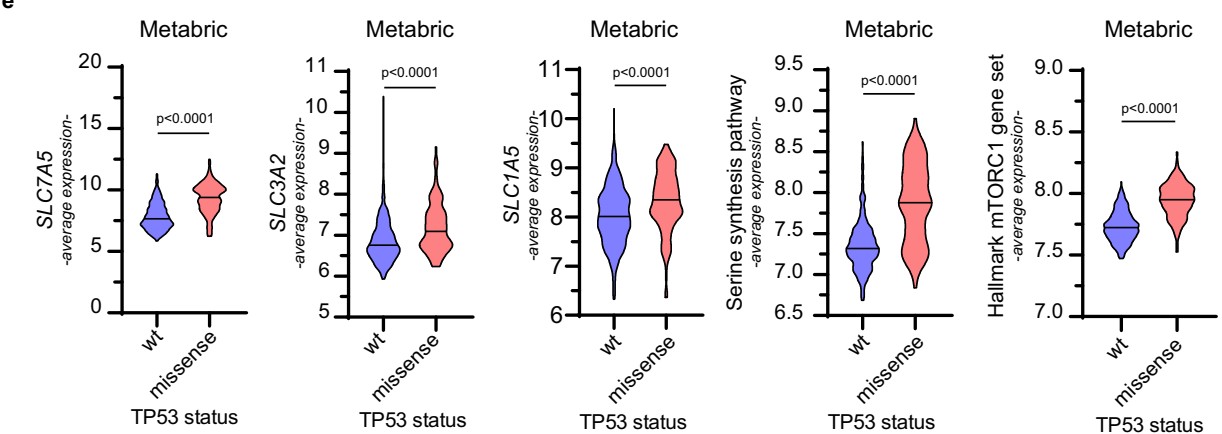

impacts mutp53 accumulation and activation in cells exposed to low AA concentration. Stabilization of mutp53 protein in cancer cells has been shown to involve histone deacetylase 6 (HDAC6)-mediated Hsp90 deacetylation, which protects mutp53 from murine double minute 2 (MDM2)-dependent degradation[22,53]. Remarkably, the accumulation of mutp53 in low AA was prevented by pharmacological inhibition of either HDAC6 (SAHA) or HSP90 (17-AAG) (Suppl. Fig. 4h).

Altogether, this evidence suggests that in cancer cells experiencing a reduction in AA availability, mutp53 is stabilized and in turn supports tumor cell proliferation.

**Fig. 2 | BCs with mutp53 express higher levels of SSP enzymes, LAT1, and mTORC1 activation. a** Quantification of tumor volume in mice injected with indicated 4T1 TetOn inducible clones at day 24; $n = 4$/condition. **b** qRT-PCR analysis of the indicated genes in 4T1 TetOn clones used in (**a**); $n = 4$/condition. **c** Left panel: western blot analysis of the indicated proteins in lysates of tumor samples from (**a**). HSP90 was used as loading control. Right panel: quantification of depicted western blot bands expressed as percentage of S6RP pS240-244 relative to S6RP. In vivo data (**a**) are shown as mean ± SE. In scatter dot plots, each dot represents one mouse. Graph bars in **b** represent mean of $n = 4$ mice per group. Graph bars in **c** represent mean ± s.d. $n = 4$ mice per group. **d** Upper panels: representative images

of immunohistochemical (IHC) analysis of 10 breast cancer samples stratified on the basis of p53 protein levels (high and low) ($n = 5$ for each condition). Samples were stained with anti-p53, anti-PSAT1, anti-LAT1, anti-CD98hc, and anti-Phospho-4EBP1 antibodies. CD98hc is encoded by *SLC3A2* and LAT1 by *SLC7A5*. Original magnification, x200. Scale bar, 100 µm. Lower panels: quantitative analyses of IHC markers analyzed. **e** Average expression levels of *SLC7A5*, *SLC3A2*, *SLC1A5*, SSP genes (i.e., *PHGDH*, *PSAT1*, and *PSPH*), and a gene set of mTORC1 activation in human breast cancer samples of the Metabric dataset ($n = 701$) classified according to p53 status (wt and missense *TP53* mutations). Two-tailed Student's *t*-test. Source data are provided as a Source Data file.

## Mutant p53 rewires AA metabolism upon AA restriction

To evaluate the impact of mutp53 on the metabolic profile of BC cells, we then performed a targeted metabolomic analysis on MDA-MB-231 cells upon AA restriction and mutp53 silencing. As shown in Fig. 4, AA restriction and mutp53 knockdown dramatically impacted the metabolome. Principal Component Analysis (PCA) clearly distinguished the four different experimental conditions (Suppl. Fig. 5a). Down-regulation of mutp53 (Suppl. Fig. 5b) led to alterations in different metabolic pathways when cells were grown in CM or in low AA: in particular, mutp53 ablation resulted in changes in pathways related to AA metabolism including Ser, Gly, Leu, Iso, Val, Met metabolism in cells exposed to AA restriction (Suppl. Fig. 5c).

Analysis of individual metabolic intermediates highlighted 51 significantly modulated metabolites (FDR < 0.05) (Suppl. Table 5). In CM, silencing of mutp53 significantly reduced the levels of Asp (Fig. 4a, b), while other AAs were unaffected (Fig. 4a, c–h, Suppl. Fig. 5d, e).

A different metabolic scenario emerged when mutp53 was silenced in cells grown in low AA. Except for Gln and Glu (Suppl. Fig. 5d, e), which increased upon mutp53 silencing, a significant reduction was observed for several AAs, including Asp, Phe, Met, Ile, Asn, and the overall pool of EAAs (Fig. 4b, e–i).

In Fig. 1a we showed that silencing of mutp53 in cells grown in CM led to decreased serine biosynthesis. However, when we assessed serine levels by steady-state metabolomic analysis (Fig. 4d), we did not observe relevant changes, likely due to serine intake by transporters. Under low AA conditions, serine levels declined, and this trend was further enhanced upon mutp53 silencing (Fig. 4d). We also evaluated the impact of mutp53 on Ser/Gly biosynthesis by conducting glucose tracing experiments under the aforementioned conditions. As expected, low AA availability boosts glucose-derived serine and glycine production, an effect that was completely abolished by mutp53 downregulation (Fig. 4j).

Having demonstrated that mutp53 potentiates SGOC metabolism and the import of EAAs via LAT1, we sought to identify which among the products of these pathways (i.e., nucleotides, glutathione and S-adenosylmethionine (SAM)) was critical for the survival and proliferation advantage conferred by mutp53 to cancer cells under AA restriction (Fig. 1g)[10,54]. To this aim, we supplemented MDA-MB-231 cells, silenced for mutp53 and cultured in low AA, with either nucleotides (dNTPs), SAM or N-acetylcysteine (NAC). As shown in Fig. 4k, the addition of dNTPs completely rescued tumor cell proliferation in low AA. Also NAC caused a slight improvement in cell proliferation rate, although less than addition of dNTPs. Conversely, SAM administration did not exhibit any effect on cell proliferation. Under low AA compared to CM, the oxidative branch of PPP was upregulated as demonstrated by higher levels of 6-phosphogluconate, ribose-5-phosphate/xylulose-5-phosphate/ribulose-5-phosphate (three different isomers that we measured together and from now on named ribose-5-phosphate) (Suppl. Fig. 5f, g). On the other hand, in cells silenced for mutp53 in low AA medium compared to control, ribose-5-phosphate levels were decreased as well as those of NADPH (Suppl. Fig. 5g, h), a reducing cofactor produced by the PPP oxidative phase with a key role in maintaining redox balance. In low AA condition, we detected an

increased amount of oxidized glutathione (GSSG) in cells silenced for mutp53, likely due to the reduced levels of NADPH (Suppl. Fig. 5i). These findings suggest that mutp53 is necessary for maintaining the redox status of tumor cells. Accordingly, the rescue of proliferation rate observed only with dNTPs administration in mutp53-silenced cells under low AA conditions is likely due to the bypass of both AAs utilization and the PPP oxidative phase, pathways that are suppressed when mutp53 is silenced.

Notably, reduced nucleotide synthesis and alteration of tumor redox balance, as well as of SGOC metabolism in general, can result in increased DNA damage and reduced tumor growth[13,17,55–57]. Indeed, cells deprived of AAs exhibited increased DNA damage that was further enhanced upon mutp53 knockdown (Suppl. Fig. 5j).

## Mutant p53 induces an AA metabolic transcriptional program in cancer cells exposed to low AA

To unravel the mechanisms underlying mutp53-dependent metabolic alterations, we performed RNA-seq upon mutp53 knockdown in MDA-MB-231 cells grown in low AA. In control cells, the gene sets that were most enriched upon AA restriction were related to the canonical transcriptional response to AA scarcity, including metabolism of AAs and derivatives, tRNA aminoacylation and AA transport (Fig. 5a and Suppl. Fig. 6a). Indeed, most upregulated transcripts encode for enzymes and transporters that fuel SGOC, including *PSAT1*, *SLC1A4*, and *CTH*, and others involved in nutrient stress response, including Growth/differentiation factor 15 (*GDF15*)[58] and Phosphoenolpyruvate Carboxykinase 2 (*PCK2*)[59,60]. Interestingly, mutp53 knockdown significantly reduced the expression of all major gene sets enriched upon AA restriction, along with a cell cycle gene set (Fig. 5a and Suppl. Fig. 6b). The volcano plot in Fig. 5b highlights that mRNAs most downregulated upon mutp53 silencing were those encoding for SGOC enzymes, including *PHGDH*, *PSAT1*, *MTHFD2*, *NNMT*, and *SHMT2*, and for the AA transporters *SLC1A4*, *SLC1A5*, *SLC3A2*, and *SLC7A5*. These findings suggest that mutp53 plays a pivotal role in sustaining cancer cells' adaptive response to low AA conditions. Of note, this effect could not be ascribed to a reduced activation of GCN2, as increased phosphorylation of eIF2α in low AA occurred irrespective of mutp53 expression (Suppl. Fig. 6c).

Intriguingly, some of the genes induced by mutp53 under low AA were also regulated by mutp53 in complete medium, although to a lesser extent (Fig. 1b, c and Suppl. Fig. 6i). This suggests that AA scarcity may promote the mutp53-dependent induction of AA metabolism genes that, in turn, supports survival and proliferation of cancer cells. Indeed, the significant upregulation of genes encoding SSP enzymes and *SLC7A5*/*SLC3A2* EAAs transporter in mutp53- expressing cells grown in low AA, was almost completely abolished upon mutp53 knockdown (Fig. 5c and Suppl. Fig. 6d, e). As expected, as a consequence of the canonical AA stress response we observed increased expression of SSP, SLC7A5/SLC3A2 genes and PSAT1, LAT1/CD98hc protein levels upon growth in low AA also in wtp53-expressing BC cells. This effect however was not dependent on wtp53, and wtp53 protein levels were not affected by growth in low AA (Suppl. Fig. 6f–h). To confirm these findings, we employed 3D cultures of DCIS cells expressing doxycycline-inducible mutp53. In this setting, we observed

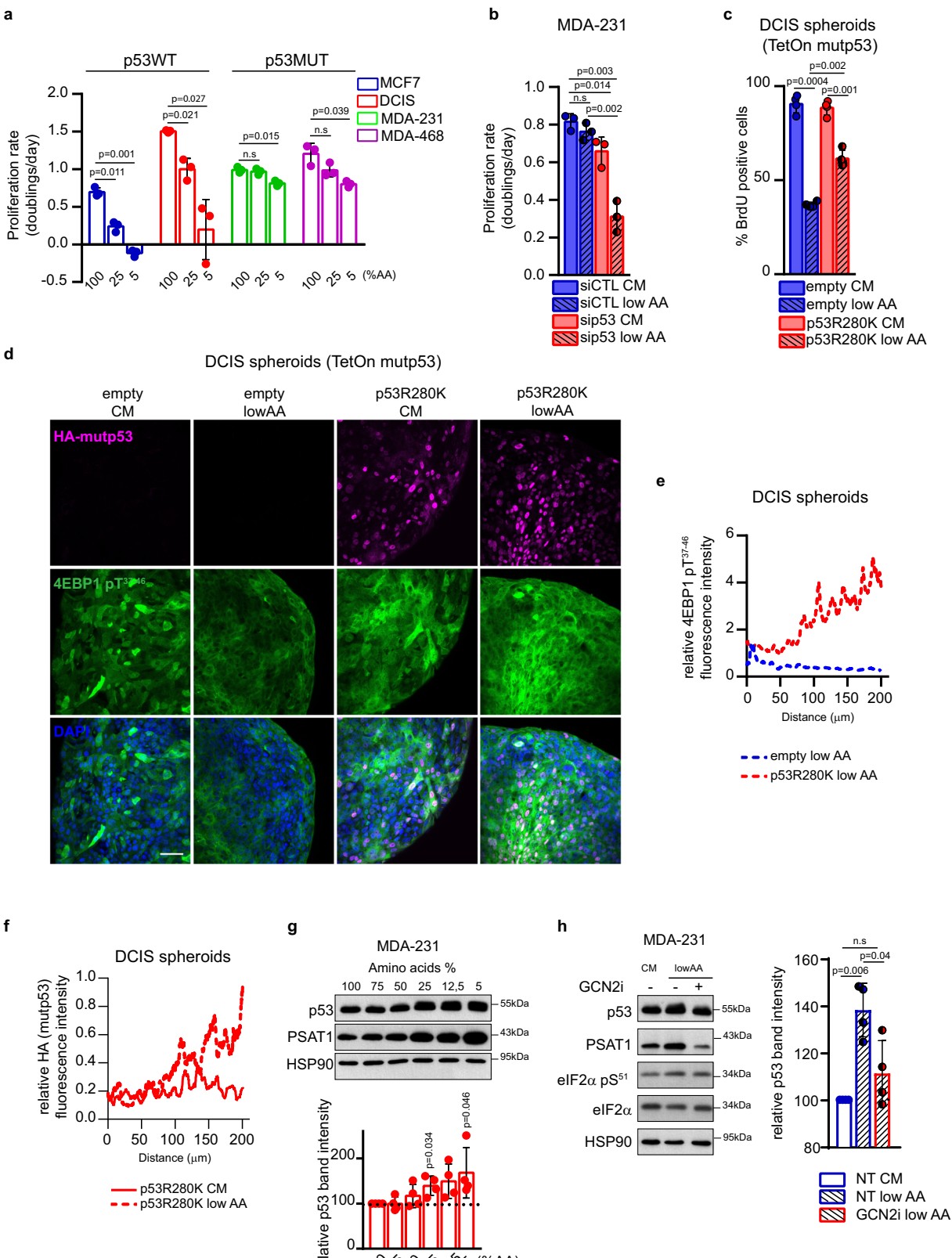

significant upregulation of *SLC7A5, SLC3A2* and SSP genes in mutp53-expressing spheroids exposed to low AA, while induction of these genes was much more limited in samples expressing only wtp53 (Fig. 5d).

Furthermore, under low AA conditions mutp53 upregulated some AA metabolism genes that were either not induced or even repressed

by mutp53 itself in complete medium, i.e., *CTH*, and *SLC1A4*, while other known mutp53 target genes (such as *BUB1, CCNE2, PSMA1, PSMB1* and others) were not induced by low AA[61,62] (Fig. 1b, Fig. 5b and Suppl. Fig. 6i). This suggests that AA restriction potentiates a specific subset of the mutp53 transcriptional repertoire. Notably, in BC patients this mutp53-dependent AA metabolism signature (Suppl. Table 6) was

**Fig. 3 | Mutp53 enables cancer cells to cope with AA restriction. a** Proliferation rate (doubling times/day) of indicated cell lines in medium containing 100%, 25%, and 5% of AAs for 3 days; $n = 3$ biological replicates. **b** Proliferation rate (doublings /day) of MDA-MB 231 cells cultured in complete medium (CM) or medium containing 25% of AAs (low AA) upon silencing of p53 for 3 days; $n = 3$ biological replicates. **c** BrdU incorporation analysis in indicated MCF10DCIS.COM TetOn inducible spheroids grown in complete medium (CM) or medium containing 25% of AAs (low AA), in presence of doxycycline 1 μg/mL for 72 h. BrdU was added 12 h before harvesting. The percentage of BrdU positive cells in 900 nuclei/spheroid is shown; $n = 4$ biological replicates. **d** Immunofluorescence analysis of HA-mutp53 and p4EBP1 in MCF10DCIS.COM spheroids maintained for 72 h in complete medium (CM) or medium containing 25% of AAs (low AA) upon overexpression of mutp53 R280K (in presence of doxycycline 1 μg/mL for 72 h); $n = 4$ biological

replicates. Scale bar 40 μm. **e** Quantification of the relative fluorescence intensity profile of p4EBP1, normalized according to nuclei staining (DAPI). **f.** Quantification of the relative fluorescence intensity profile of HA-mutp53, normalized according to nuclei staining (DAPI). **g** Upper panel: western blot analysis of the indicated proteins in MDA-MB 231 cells grown for 72 h in the indicated percentage of AAs. Lower panel: quantification of p53 levels relative to HSP90 in western blot; $n = 4$ **h** Left panel: western blot analysis of the indicated proteins in MDA-MB 231 cells grown for 72 h in complete medium (CM) and in medium containing 25% of AAs (low AA) with (+) or without (-) GCN2i 1 μM for 72 h. Right panel: quantification of p53 levels relative to HSP90 in western blot; $n = 4$ biological replicates. Graph bars represent mean±s.d. Two-tailed Student's *t*-test. Source data are provided as a Source Data file.

associated with missense mutp53 status and correlated with poor prognosis (Fig. 5e, f and Suppl. Fig. 6j).

In solid tumors, the presence of a rigid fibrotic stroma can lead to inefficient perfusion and increased interstitial pressure, causing fluctuations in AA availability[63–65]. Notably, ECM stiffening also leads to stabilization and activation of mutp53[22]. Therefore, we investigated whether ECM-derived mechanical cues could sustain mutp53-dependent induction of AA metabolism, particularly under AA scarcity. To explore this, we cultured MDA-MB-231 cells on fibronectin-coated hydrogels with varying elastic moduli. As shown in Suppl. Fig. 7a, b, ECM stiffening elevated mutp53 protein levels and resulted in a p53-dependent increase in the expression of *SLC7A5* and *PSAT1*. Conversely, BC cells grown on plastic (a hyperstiff substrate) exhibited reduced mutp53 protein levels and decreased transcription of *SLC7A5* and *PSAT1* genes when treated with mechanosignaling inhibitors that target focal adhesion signaling (e.g., FAK inhibitor PF573228, Src family inhibitor Dasatinib)[66,67] and actin polymerization (e.g., Cytochalasin D and Latrunculin A)[68] (Suppl. Fig. 7c, d). Consistently, mutp53-bearing tumors characterized by high LAT1 and PSAT1 levels (Fig. 2d) displayed collagen fibers embedding tumor foci (Suppl. Fig. 7e).

These results led us to hypothesize that mechanosignaling could be required for mutp53 stabilization in low AA. To further investigate this aspect, MDA-MB-231 cells grown on plastic and exposed to low AA, were treated with mechanosignaling inhibitors, including Dasatinib, PF573228 or Cytochalasin D. We observed that the upregulation of mutp53 induced by low AA was prevented by these treatments (Suppl. Fig. 7f). Moreover, treatment with Dasatinib and PF573228 abrogated the transcriptional activation of *SLC7A5/SLC3A2* and SSP enzymes induced by AA restriction and enhanced the inhibitory effect of low AA on cancer cell proliferation (Fig. 5g, h).

These data suggest that, in BC cells expressing mutp53, signaling along mechanotransduction pathways is crucial for activating a specific mutp53-dependent transcriptional program, providing a proliferation/survival advantage under AA scarcity.

### Inhibition of serine synthesis and EAAs intake reduces mutant p53-dependent BC growth in vivo

To investigate the significance of SSP and LAT1-mediated AAs transport for the oncogenic function of mutp53 in BC cells, we treated MDA-MB 468 mutp53-expressing cells cultured in low AA, with NCT503 (a specific inhibitor of PHGDH, a key enzyme in the SSP), and JPH203 (a LAT1 transporter inhibitor)[40]. Each treatment reduced cell growth in low AA, and combined inhibition of both PHGDH and LAT1 (N/J) showed a stronger effect (Suppl. Fig. 8a). Importantly, the proliferation/survival advantage of mutp53-expressing DCIS and 4T1 cells in low AA was abolished by inhibition of either PHGDH, LAT1 or their combination (Fig. 6a, b).

Furthermore, the effect of these inhibitors on mutp53-dependent tumorigenesis was investigated in vivo. 4T1 cells expressing

doxycycline-inducible HA-tagged mutp53 variants (R175H or R280K) or control vector were injected into the mammary fat pad of syngeneic mice, and mutp53 expression was induced by doxycycline administration. Expression of either mutp53[R280K] or mutp53[R175H] significantly increased tumor growth (Fig. 6c–e and Suppl. Fig. 8b). Once tumors were palpable, mice were administered combined NCT503/JPH203 treatment or placebo. The combined treatment with serine synthesis and LAT1 inhibitors had minimal impact on the growth of control (p53-null) tumors, whereas it markedly reduced both the volume and weight of tumors expressing mutp53 variants (Fig. 6d, e). Importantly, the growth advantage conferred by mutp53 expression in vivo was completely abolished by NCT503/JPH203 treatment (Fig. 6c–e). The combination treatment did not show any general toxicity as judged by body weight evaluation (Suppl. Fig. 8c). Moreover, mutp53-expressing tumors treated with NCT503/JPH203 exhibited increased γH2AX *foci* and expression of the apoptotic marker cleaved caspase 3 (Suppl. Fig. 8d, e), suggesting that combined inhibition of serine synthesis and LAT1-mediated AA intake may increase both DNA damage response and cell death, thus contributing to tumor reduction.

Altogether, these results provide evidence that targeting the SSP and LAT1 transporter, which mediates EAAs uptake, abolishes the oncogenic function of mutp53 in vitro and inhibits tumor growth in vivo.

### Patient-derived tumor organoids expressing mutant p53 are vulnerable to SSP/LAT1 or mechanosignaling inhibition under AA restriction

Patient-derived organoids (PDOs) recapitulate the epithelial architecture and the spatial and nutrient heterogeneity of their tumor of origin[69]. We sought to employ these ex-vivo models to confirm the ability of mutp53 to confer growth/survival advantages to cancer tissues in conditions of AA scarcity, and to test potential tumor vulnerabilities associated with this metabolic dependency. PDOs were generated from seven tumor samples representing different BC subtypes, with three expressing wtp53 and four harboring distinct missense *TP53* mutations (Fig. 7a). These PDOs were cultured in media with progressively reduced AAs supply (100%, 25% and 5% relative AAs content).

We observed that PDOs expressing wtp53 displayed significantly decreased viability in low AA conditions, whereas all mutp53-expressing PDOs were still able to proliferate (Fig. 7b), thus reflecting the varied sensitivity to nutrient stress observed among BC cells expressing different p53 variants. Accordingly, BC tissues from which mutp53-expressing PDOs were derived, exhibited higher levels of PSAT1 and LAT1 compared to those expressing wtp53 (Suppl. Fig. 9a).

We next evaluated the impact of inhibiting either the SSP and EAAs transporter or mechanosignaling on the growth of PDOs upon reduced AA availability. Strikingly, treatment with NCT503 and JPH203, either as single agents or in combination (N/J), as well as with

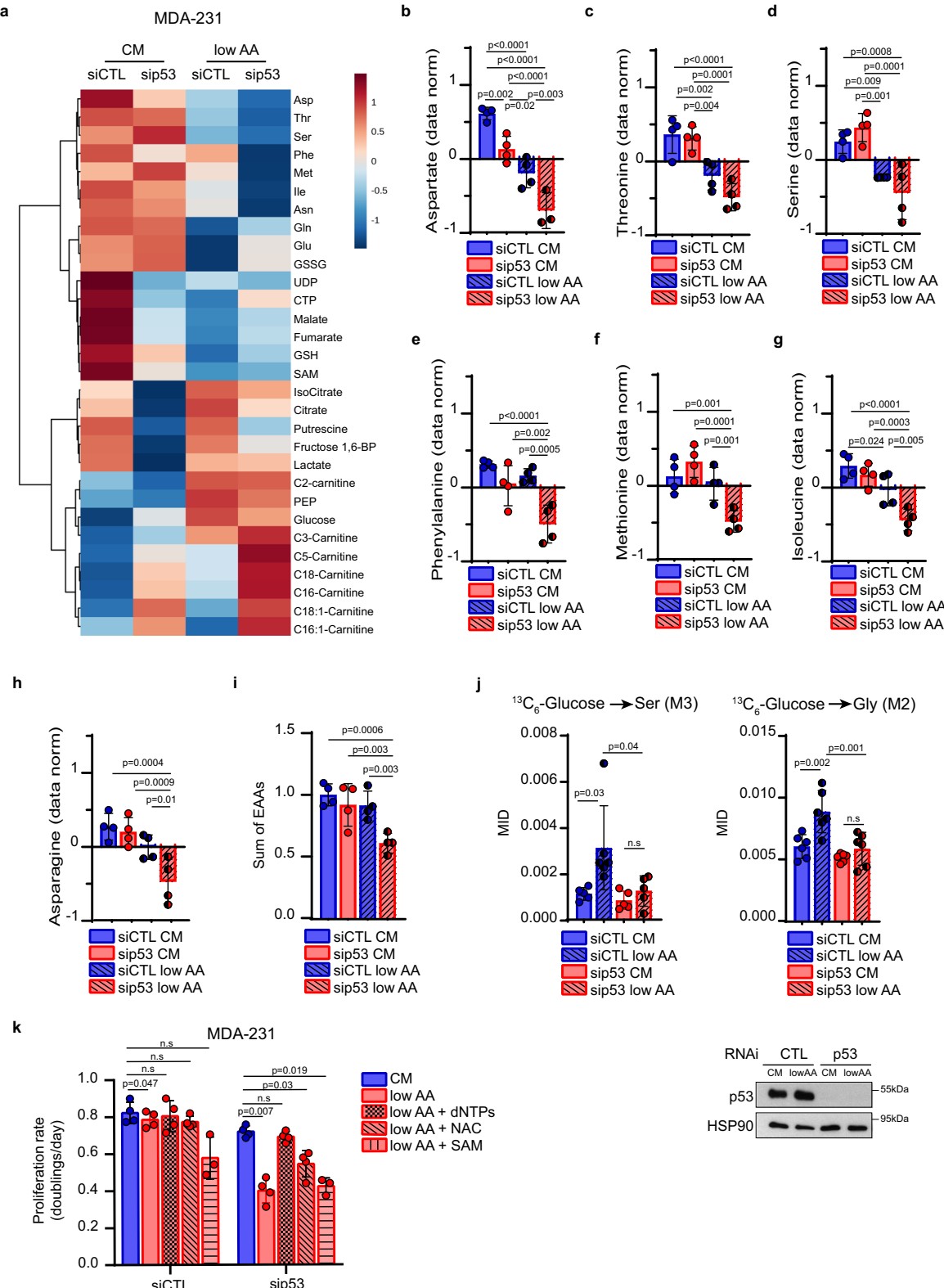

Dasatinib, significantly decreased the viability of all mutp53-expressing PDOs (Fig. 7c, d). Importantly, these treatments did not further decrease the viability of PDOs harboring wtp53 when cultured under low AA conditions (Fig. 7c, d). These findings underscore that mutp53-bearing BC PDOs exhibit a proliferative and survival advantage under conditions of AA restriction, which represents a potential vulnerability.

## Discussion

Overexpression of SGOC enzymes and EAAs transporters is frequent in several tumors and associates with poor prognosis, making them promising targets for cancer therapies[3,35,38,70]. Cancer cells have a high demand for serine and tumors arising in tissues with low serine supply, such as breast, gain a growth advantage through increased serine

**Fig. 4 | Mutp53 reprograms AA metabolism in AA restriction to sustain proliferation. a** Heatmap of top 30 significantly regulated metabolites analyzed by steady-state metabolomic analysis in MDA-MB 231 cells cultured for 3 days in complete medium (CM) or medium containing 25% of AAs (low AA) upon control (siCTL) or p53 (sip53) silencing. Each column represents the average of each experimental group; *n* = 4 independent replicates (one single experiment). **b**–**h** Histograms showing the normalized abundance of top 7 regulated AAs emerged from **a**. *n* = 4 independent replicates (one single experiment). **i** Histogram showing the sum of EAAs (normalized values of Val, Thr, Leu, Ile, Met, His, Phe, Trp, Lys) from LC-MS analysis. *n* = 4 independent replicates (one single experiment). **j** Upper panels: mass isotopomer distribution (MID) of serine M3 (left) and glycine M2

(right) from [U-$^{13}$C$_6$]-Glucose in MDA-MB 231 cells under above-described conditions. For labeling experiments, [U-$^{13}$C$_6$]-Glucose was added in the medium 24 h before harvesting. Lower panel: western blot analysis of p53 levels in the above-described condition. HSP90 was used as loading control; *n* = 6 biological replicates. **k** Proliferation rate (doublings/day) of MDA-MB 231 cells cultured in complete medium (CM) or medium containing 25% of AAs (low AA) upon silencing of p53 for 3 days and, where indicated, supplemented with dNTPs 100 µM, N-Acetylcysteine (NAC) 1.25 mM and S-adenosylmethionine (SAM) 200 µM; *n* = 4 (*n* = 3 in low AA + SAM) biological replicates. Two-tailed Student's *t*-test or Ordinary one-way ANOVA test (Fisher's LSD). Source data are provided as a Source Data file.

synthesis[12], which can be achieved by increasing either glycolytic intermediates or directly promoting SSP.

In this study we found that, in BC, mutp53, but not wtp53, directly enhances the expression of all SSP enzymes, boosting de novo serine synthesis. This effect is more pronounced under conditions of low AA availability, promoting BC cells proliferation. These observations suggest that the tissue context and the type of nutritional stress can influence mutp53-dependent adaptive responses. Mutp53 is known to promote glucose intake[26], which also feeds serine synthesis through glycolysis. When glucose is scarce, adequate Ser/Gly production is guaranteed by the gluconeogenic enzyme PCK2[71]. Interestingly, we found that mutp53 promotes the expression of PCK2, which may boost serine production by ensuring substrates availability and shunting carbon units from glycolysis into SSP. Furthermore, in BC, we observed a correlation between missense mutp53 and increased expression of the two major serine transporters ASCT1 (*SLC1A4*) and ASCT2 (*SLC1A5*), frequently overexpressed in cancer[34,72]. Therefore, it is conceivable that mutp53-bearing tumors maintain high serine levels, by increasing both intracellular transport and synthesis of serine, thereby sustaining proliferation and redox balance.

On the other hand, the availability of EAAs in cancer cells relies on their efficient uptake from the environment. We found that mutp53 promotes the influx of EAAs, by sustaining the expression of both LAT1 transporter and of its plasma membrane anchor CD98hc. LAT1/CD98hc acts as an antiporter mediating EAAs influx principally in exchange for glutamine, which is imported by ASCT2. Interestingly, we observed significantly higher expression of LAT1/CD98hc and ASCT2 in breast tissues bearing mutp53 compared to wtp53 tumors. Hence by increasing the expression of these transporters, mutp53 could maximize the influx of EAAs, including leucine and methionine, which are crucial for mTORC1 activation and one-carbon metabolism.

Therefore, our results suggest that mutp53 potentiates two interconnected branches of AA metabolism[17,73,74], i.e., SSP and EAAs intake (Suppl. Fig. 9b), which converge to fuel one-carbon cycle, a central hub for nucleotides synthesis, redox balance and epigenetic regulation, thus ensuring metabolic adaptation to nutrient fluctuations[54]. Interestingly, we observed that, under low AAs availability, the GCN2-dependent AA stress response leads to the accumulation of mutp53. In turn mutp53 activates a specific transcriptional program and rewires cancer cell metabolic profile, guaranteeing high intracellular levels of AAs to provide nucleotides and antioxidant agents, crucial for DNA synthesis, replication and repair. Indeed, under low AAs, mutp53 supports nucleotides incorporation and limits DNA damage, conferring a proliferative advantage to BC cells. The AA metabolism genes controlled by mutp53 have been reported as targets of major transcription regulators, including MYC, YAP and NRF2, known to bind mutp53[75–77] and our data suggest that mutp53 cooperates with MYC to induce the transcription of AA metabolism genes.

Nutrient availability in tumors is influenced by both anatomic site and tissue architecture. For example, a tumor-constraining stiff ECM can concomitantly contribute to nutrient limitation and promote cancer progression by favoring metabolic reprogramming and activating mechanosensitive oncogenes, including mutp53[22,78–80]. We

found that ECM mechanical cues are necessary to enable a mutp53-dependent response to AA scarcity and that either targeting mechanosignaling upstream of mutp53 activation or inhibiting its downstream AA metabolic pathways, could effectively suppress mutp53 oncogenic activities in BC.

Dietary or pharmacological interventions aimed at interfering with either serine or LAT1-transported AA metabolism have been proposed to reduce tumor growth[17,81]. However, prior studies have shown the inefficacy of single-agent metabolic therapies, due to emerging compensatory metabolic responses. For example, inhibition of PHGDH (NCT503) in osteosarcoma increased LAT1/CD98hc expression and augmented intracellular BCAAs with consequent mTORC1 activation and increased survival[74]. Similarly, BC cells resistant to LAT1 inhibition (JPH203) displayed enhanced expression of SGOC pathway genes[82]. Our data in BC cells, mouse models, and PDOs demonstrate that combined administration of PHGDH and LAT1 inhibitors significantly blunts the growth advantage of mutp53-bearing tumors, especially under AA limitation. These findings suggest that mutp53-bearing BCs become dependent to these metabolic pathways. Moreover, we provided a proof of concept that inhibiting mechanosignaling using Dasatinib, a Src family inhibitor currently in phase II clinical trial for advanced BC[83,84], could be an effective strategy to halt the growth of mutp53-BCs under AA scarcity (Suppl. Fig. 9b).

*TP53* mutations are the most common genetic lesions reported during BC brain metastasis[85,86] which relies on glutamine and BCAAs, highly abundant in the brain, while depending on efficient SSP, due to severe depletion of serine and glycine in this tissue[13,87]. It is tempting to speculate that the ability of mutp53 to promote EAAs intake and Ser/Gly synthesis may also endow BC cells with an advantage during colonization of secondary organs, such as the brain, where the availability of these AAs is challenging.

All in all, our study supports the notion that missense mutp53 acts as a context-activated oncogene, favoring cancer cells adaptation to tumor-related stresses. This implies that during all stages of tumor evolution, tissue environmental conditions, including AAs fluctuations and stiff ECM, may act as critical selectors for mutp53-expressing tumor cells, which display metastatic and chemo-resistant features.

## Methods
### Cell lines
MDA-MB-231, MDA-MB-468, SUM149PT, MCF7, and MCF-10A.DCIS.COM are human breast cancer cell lines. MDA-MB-231 and MDA-MB-468 were cultured in Dulbecco's Modified Eagle's Medium (DMEM, LONZA) supplemented with 10% Fetal Bovine Serum (FBS) 100 U/mL penicillin and 10 µg/mL streptomycin. SUM149PT were cultured in Dulbecco's Modified Eagle's Medium (DMEM)/F12 (LONZA) (1:1) supplemented with 10% Fetal Bovine Serum (FBS), 100 U/mL penicillin, 10 µg/mL streptomycin, 1%. MCF7 were cultured in Eagle's Minimum Essential Medium (EMEM, Sigma) supplemented with Fetal Bovine Serum (FBS), 100 U/mL penicillin, 10 µg/mL streptomycin, 1% Minimum essential medium non-essential amino acids (MEM NEAA), and 10 µg/ml recombinant human insulin. MCF-10A.DCIS.COM cells were cultured in Dulbecco's Modified Eagle's Medium (DMEM)/F12

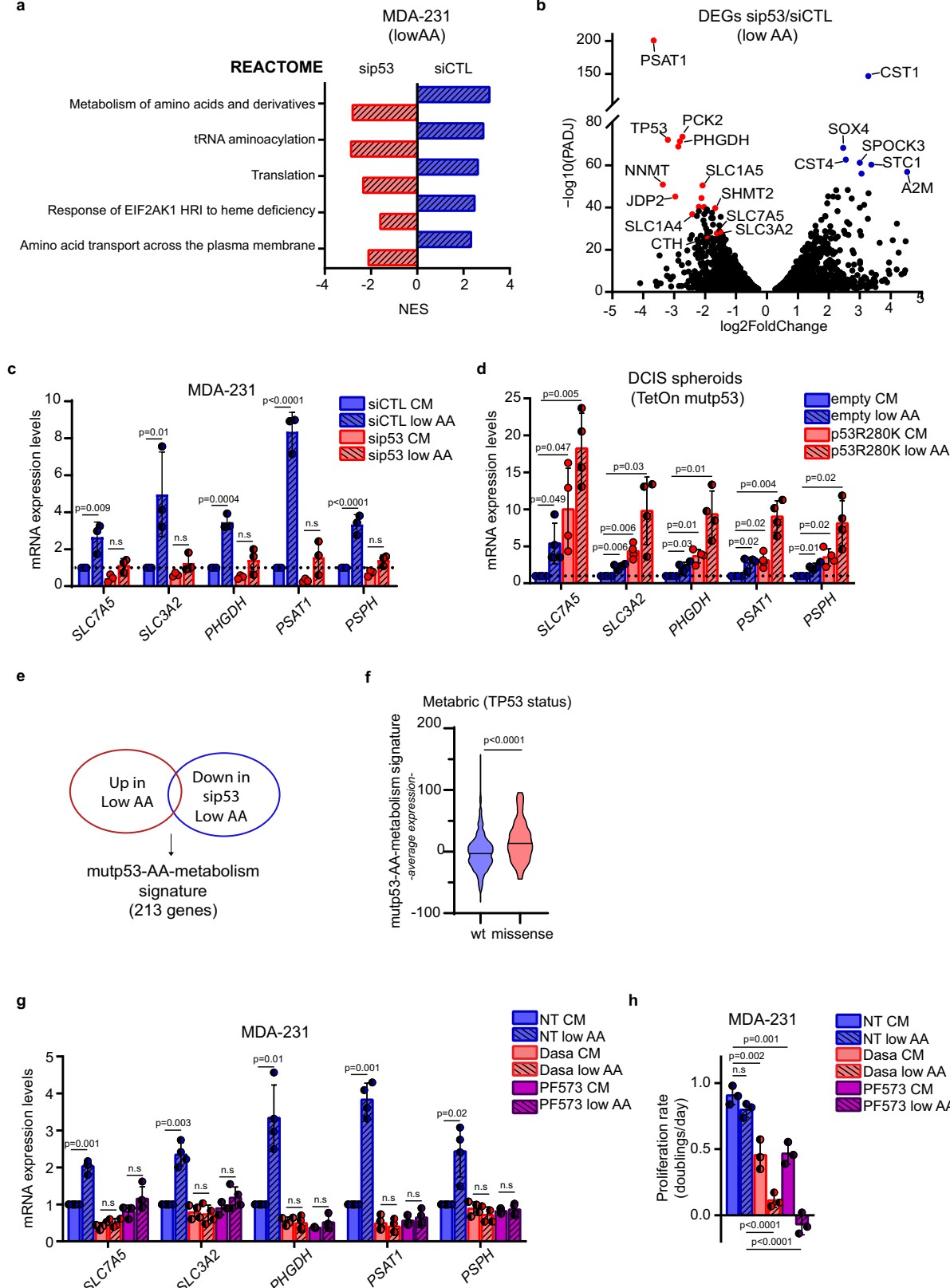

(LONZA) (1:1) supplemented with 5% Horse Serum (HS), 100 U/mL penicillin and 10 μg/mL streptomycin, 20 ng/ml recombinant human epidermal growth factor (EGF), 10 μg/ml recombinant human insulin, 500 ng/ml hydrocortisone. HEK-293T is a human embryonic kidney cells cultured in Dulbecco's Modified Eagle's Medium (DMEM, LONZA)

supplemented with 10% Fetal Bovine Serum (FBS) 100 U/mL penicillin and 10 μg/mL streptomycin. 4T1 is a mouse breast cancer cell line cultured in Dulbecco's Modified Eagle's Medium (DMEM)/F12 (LONZA) (1:1) supplemented with 10% Fetal Bovine Serum (FBS), 100 U/mL penicillin, 10 μg/mL streptomycin and Glutamax 1%.

**Fig. 5 | Mutp53 controls a specific AA metabolic transcriptional program in AA restriction. a**. Blue bars: reactome gene sets significantly enriched (padj value < 0.05) in control-silencing MDA-MB-231 cells (siCTL) cultured in low AA vs complete medium. Red bars: reactome gene sets significantly enriched (padj value < 0.05) in MDA-MB-231 cells cultured in low AA upon p53 silencing (sip53) vs low AA upon control silencing. The positive and negative normalized enrichment score (NES) indicates the degree to which Reactome gene sets are overrepresented in the conditions described above. $n = 2$ biological replicates (GSE214494). **b** Volcano plot of genes significantly upregulated (blue dots) and downregulated (red dots) (padj value < 0.05) in MDA-MB-231 cells cultured in low AA upon p53 silencing vs low AA upon control silencing (sip53/siCTL). $n = 2$ biological replicates. **c** qRT-PCR of indicated genes in MDA-MB 231 cells cultivated for 72 h in complete medium (CM) or medium containing 25% of AAs (low AA) upon silencing of p53. mRNA levels relative to control condition (dotted line) are shown; $n = 3$ biological replicates. **d** qRT-PCR of indicated genes in MCF10DCIS.COM TetOn inducible spheroids grown in complete medium (CM) or medium containing 25% of AAs (low AA), in presence of doxycycline 1 μg/mL for 72 h. mRNA levels relative to control condition (dotted line) are shown; $n = 4$ biological replicates. **e** Venn Diagram depicting common genes significantly upregulated (padj value < 0.05) in MDA-MB-231 siCTL cells when cultured in low AA and significantly downregulated upon silencing of p53 in low AA. **f** Average expression levels of mutp53-AA-metabolism signature in human BC samples of the Metabric dataset ($n = 701$) classified according to p53 status. **g** qRT-PCR of the indicated genes in MDA-MB 231 cells cultured in complete medium (CM) or medium containing 25% AAs (low AA) and treated with DMSO (NT), Dasatinib (Dasa) 0,5 μM or PF573228 (PF573) 10 μM for 72 h. mRNA levels relative to untreated condition in complete medium (NT CM) are shown; $n = 4$ biological replicates. **h** Proliferation rate (doublings/day) of MDA-MB 231 cells cultured as in **g**; $n = 3$ biological replicates. Graph bars represent mean±s.d. Two-tailed Student's $t$-test or Ordinary one-way ANOVA test or Wilcoxon Rank Sum test or Wald test (pval) and bonferroni correction (padj). Cutoff padj<0.05. Source data are provided as a Source Data file.

Human cell lines are from ATCC or other laboratories cooperating on the project. Cells were tested for mycoplasma contamination with negative results.

## Breast cancer organoid culture

For organoid generation, tissues were processed as described in Chen et al., 2021[88]. Briefly, tissues were mechanically processed until small portions (1 mm³) were obtained. Then they were enzymatically digested for 2–6 h at 37 °C with a digestion buffer containing Collagenase type III 300 U mL$^{-1}$ (Worthington) Hyaluronidase 100 U mL$^{-1}$ Sigma Aldrich in Advanced DMEM/F-12 supplemented with 5% Fetal Bovine Serum (FBS), 100 U/mL penicillin and 10 μg/mL streptomycin, 10 ng/ml recombinant human epidermal growth factor (EGF), 5 μg/ml recombinant human insulin, 500 ng/ml hydrocortisone 20 ng/mL).

Isolated cell clusters were resuspended in Matrigel® (Corning) and plated in 24-well plates.

BC organoids were grown in the previously established culture conditions[89]. Briefly BC organoids were maintained in Matrigel and in Ad-DM medium supplemented with 1X Glutamax, 10 mM Hepes, 1X Penicillin/Streptomycin, 50 μg/mL Primocin, 1X B27 supplement, 5 mM Nicotinamide, 1.25mM N-Acetylcysteine, 250 ng/mL R-spondin 3, 5 nM Heregulin β−1, 100 ng/mL Noggin, 20 ng/mL FGF-10, 5 ng/mL FGF-7, 5 ng/mL EGF, 500 nM A83-01, and 500 nM SB202190. Moreover, 5 μM Y-27632 was added to culture media for the first three days of culture.

When confluency was reached, organoids were passaged by displaying and collecting Matrigel from the wells and by incubating it for 1 h at 4 °C with Cell Recovery solution (Corning). Organoids were then digested with TrypLE solution (Gibco) for 5 min at 37 °C. After enzyme neutralization and washing, organoids were resuspended in Matrigel and reseeded as above in order to allow the formation of new organoids.

Bright-field imaging of organoids was performed on an Olympus CK30 microscope.

Generation of patient-derived organoids from breast cancer was approved by institutional review board of Regina Elena National Cancer Institute and Centro di Riferimento Oncologico di Aviano (CRO), National Cancer Institute and appropriate regulatory authorities (approval no. IFO 1270/19 and IRB-06-2017 respectively). All patients signed an informed consent.

## Drug treatment of tumor organoids and viability assay

Organoids were passaged as described above and allowed to grow for 5–7 days. Organoids were then treated and after 7 days their viability was analyzed as previously described[90]. Briefly, organoids were harvested, and the smaller ones were isolated by recovering the supernatant after gentle centrifugation (16 $g$, 1 min). Then, 96-well Optiplates were coated with 20 μL of Matrigel and 20 μL of Matrigel/cells suspension were added on top (250 organoids/ well).

Medium without N-Acetylcysteine and containing the indicated drugs was then added. After 7 days of treatments, cell medium was removed and 100 μL of 1:1 CellTiter-Glo 3D Reagent (Promega)/ culture media per well were added to measure ATP as a proxy for viable cells, following the manufacturer's instructions. Luminescence reading was performed in an EnSpire® multimode plate reader (Perkin Elmer).

## Spheroid formation

For spheroid formation, a semi-confluent adherent culture of cells was detached with 0.05% Trypsin at 37 °C. After the cells have detached, the trypsin is neutralized with growth media and centrifuged. The cells are then singularized and counted. $15 \times 10^3$ singularized cells were plated on each well (three wells for each condition) of a 96-well ultra-low attachment dish (Corning® Costar® Ultra-Low Attachment Multiple Well Plate, Cat. n. CLS7007) in complete media (CM) or in 25% AA media.

## Amino acids restriction

Cells were cultivated in media with different percentages of amino acids as indicated in figure legends. These media were obtained by diluting the complete medium (depending on the cell line) with medium lacking amino acids. The latter was generated by using Dulbecco's Modified Eagle's Medium (DMEM) without amino acids and glucose or, alternatively, Hank's Balanced Salt Solution (HBSS) supplemented with MEM Vitamin solution 100X (1X, Thermo Fischer). These media were then supplemented with 100 U/mL penicillin, 10 μg/mL streptomycin, glucose, Fetal Bovine Serum (FBS) or Horse Serum (HS), and growth factors and recombinant proteins (if required) to reach the same molar concentrations of the original complete medium.

## Preparation of fibronectin-coated hydrogel matrix

50, 8, 4, or 0.5 kPa Easy Coat hydrogels (Cell guidance system) were coated with 10 μg/ml fibronectin.

## Reagents and plasmids

The following compounds and working concentrations were used: Fibronectin (10 μg/ml, Sigma Aldrich F0895), JPH203 (Selleck Chemicals S8667), NCT503 (Sigma Aldrich SML1659), Dasatinib (Selleck Chemicals S1021), PF-573228 (Selleck Chemicals S2013), Cytochalasin D (Sigma Aldrich C2618), Latrunculin-A (Cayman 10010630), SAHA Cayman 149647-78-9), 17-AAG Geldanamycin (Lc Laboratories A-6880), GCN2-IN-1 (Synonyms: A-92; MedChem Express Cat. No.: HY-100877), nucleosides dNTPs (Euroclone EMR276425), N-Acetyl-L-Cysteine (NAC) (Sigma Aldrich A9165), S-(5′-Adenosyl)-L-Methionine Iodide (SAM) (Sigma Aldrich A4377) and DMSO (Sigma Aldrich D4540). Treatments lasted as described in figure legends.

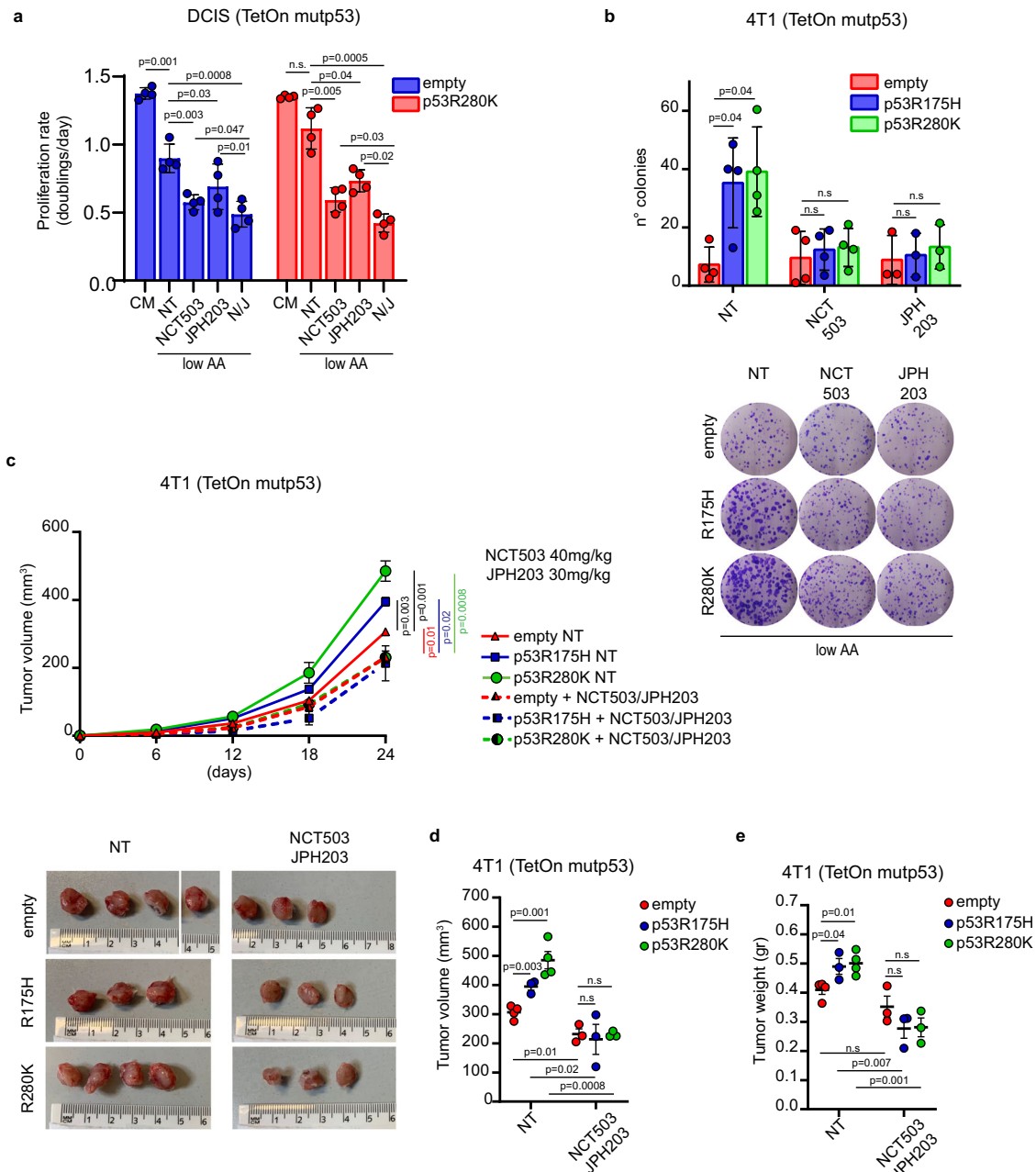

**Fig. 6 | LAT1 and SSP inhibition dampens mutp53-dependent tumor growth.**
**a** Proliferation rate (doubling times/day) of MCF10DCIS.COM TetOn inducible
clones cultured in complete medium (CM) or medium containing 25% of AAs (low
AA) in presence of doxycycline 1 μg/mL for 72 h and treated with DMSO (NT), NCT-
503 10 μM, JPH203 10 μM or combination of NCT503 10μM-JPH203 10 μM (N/J);
$n = 4$ biological replicates. **b** Upper panel: quantification of the number of colonies
formed by the indicated 4T1 TetOn inducible clones grown in medium containing
25% AAs (low AA) in presence of doxycycline 1 μg/mL, treated with DMSO (NT),
NCT-503 1 μM or JPH203 1μ. Lower panel: representative images of colonies
described above; $n = 4$ ($n = 3$ in JPH203) biological replicates. **c** Upper panel: tumor

growth rate in mice after injection of indicated 4T1 TetOn inducible clones upon
treatment with placebo (NT) or combination of NCT503/JPH203. Lower panel:
images of tumors on day 24 after injection. $n = 3$/condition except for $n = 4$ in
empty NT and p53R280K NT. **d, e** Quantification of difference in tumor volume and
weight at day 24 after injection of the above-described tumors. $n = 3$/condition
except for $n = 4$ in empty NT and p53R280K NT. Graph bars (**a, b**) represent mean
±s.d. In vivo data (**c**–**e**) are shown as mean ± SE. In scatter dot plots, each dot
represents one mouse. Two-tailed Student's $t$-test. Source data are provided as a
Source Data file.

pCW57-GFP-P2A-MCS (empty backbone) was bought from
Addgene (plasmid #89181). pCW57-GFP-HA-p53 wt, pCW57-GFP-HA-
mutp53 R175H, pCW57-GFP-HA-mutp53 R249S, pCW57-GFP-HA-
mutp53 R273K or pCW57-GFP-HA-mutp53 R280K were generated by
cloning a PCR amplified DNA fragment of the human HA-tagged *TP53*
sequence (WT or mutated) into the pCW57-GFP-P2A-MCS with MluI
and BamHI restriction sites.

**Inducible transduced cell lines**
4T1 cells overexpressing empty inducible vector, wtp53, mutp53
R175H, mutp53 R249S, mutp53 R273K, or mutp53 R280K were
obtained by lentiviral transduction with pCW57-GFP-P2A-MCS, pCW57-
GFP-HA-p53 wt, pCW57-GFP-HA-mutp53 R175H, pCW57-GFP-HA-
mutp53 R249S, pCW57-GFP-HA-mutp53 R273K or pCW57-GFP-HA-
mutp53 R280K.

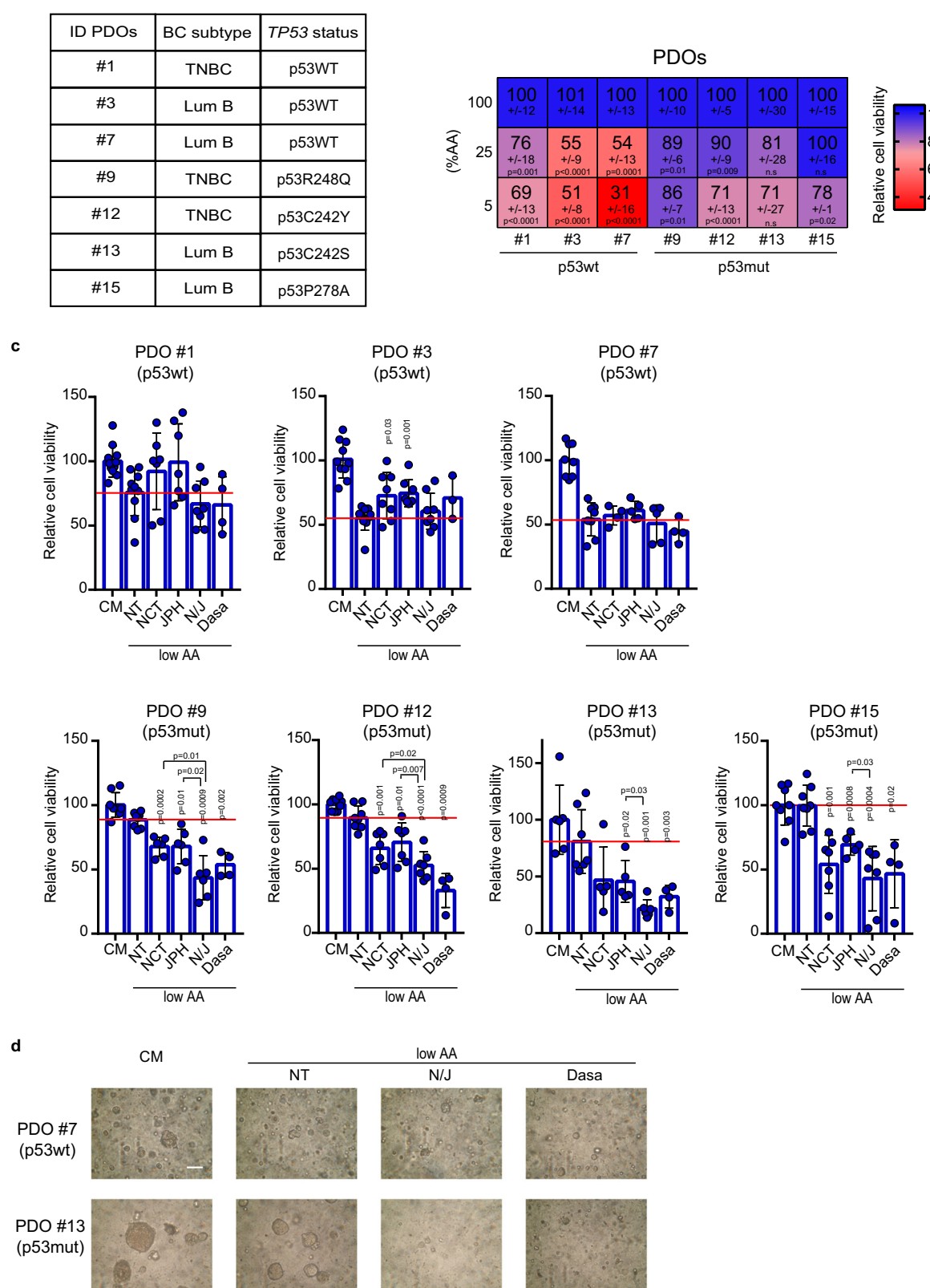

MCF10DCIS.com cells overexpressing empty inducible vector or mutp53 R280K were obtained by lentiviral transduction with pCW57-GFP-P2A-MCS or pCW57-GFP-HA-mutp53 R280K.

Infected cell populations were selected using puromycin (Sigma-Aldrich) 2,5 µg/mL for at least one week.

**Transfections**

siRNA transfections were performed with Lipofectamine RNAi-MAX (Life technologies) in an antibiotic-free medium, according to manufacturer instructions. As negative control siRNA, the Qiagen AllStars Negative Control was used. Sequences of siRNAs are reported below.

**Fig. 7 | Viability of mutp53-bearing tumor organoids in AA restriction is dampened by inhibition of SSP/LAT1 or mechanosignaling. a** Table depicting PDOs ID, BC subtype, and *TP53* status. **b** Heatmap showing relative viability percentage of patient-derived BC organoids grown in medium containing 100%, 25%, or 5% AAs. Data are normalized to complete medium (100%) and expressed as mean±s.d. of *n* = 3 experiments with at least 2 independent replicates each. *P* values are calculated vs complete medium (100%). **c** Quantification of relative viability percentage of patient-derived BC organoids grown in complete medium (CM) or in medium containing 25% AAs (low AA) and treated with DMSO (NT), NCT-503 10 μM, JPH203 10 μM, combination of NCT503 10 μM-JPH203 10 μM (N/J), Dasatinib (Dasa) 100 nM for 7 days. At the endpoint, organoids were quantified using Cell Titre Glo reagent.

Data are normalized to control treatment (CM). Graph bars represent mean±s.d. of *n* = 3 (NCT503 10 μM-JPH203 10 μM; N/J) and *n* = 2 (Dasatinib) experiments with at least 2 independent replicates each (individual replicates are shown). *P* values above each bar are calculated vs medium containing 25% AAs (low AA) and treated with DMSO (NT) (red line). *P* values calculated comparing specific conditions are indicated using square brackets. **d** Representative images of one wtp53-expressing PDO (#7) and one mutp53-expressing PDO (#13), grown in complete medium (CM) or in medium containing 25% AAs (low AA), treated with DMSO (NT), combination of NCT503 10 μM-JPH203 10 μM (N/J) or Dasatinib (Dasa). Scale bar 50 μm. Two-tailed Student's *t*-test. Source data are provided as a Source Data file.

| Oligonucleotide | Sequence | Manufacturer |
|---|---|---|
| siP53 | GACUCCAGUGGUAAUCUAC | Eurofins MWG |
| sicMYC | CGAGCUAAAACGGAGCUUU | Eurofins MWG |

For lentiviral production, low confluence HEK-293T packaging cells were transfected using PEI 2X (1 mg/ml) with the appropriate plasmids in combination with the pMD2.G and ps-PAX2 packaging vectors. After 48–72 h the viral supernatant was collected, centrifuged 5 min at 500 *g*, and filtered with a 0.45 μm low-protein binding filter to remove cellular debris. Then the virus-containing medium was added to the target cells.

## RNA extraction and qRT-PCR
Cells were harvested in Qiazol lysis reagent (Qiagen) for total RNA extraction, and contaminant DNA was removed by DNase treatment. Quantitative real-time PCR analyses were carried out on cDNAs retro-transcribed with iScript™ Advanced cDNA Synthesis Kit (Biorad 172-5038) and analyzed with BIORAD CFX96 Touch™ detection system and Biorad CFX Manager software. Experiments were performed at least three times, and each sample is the average of a technical duplicate. The quantification is based on the $2^{-\Delta\Delta Ct}$ method. Histone 3 (*H3*) was used as a reference gene in human cells while *Rpl13a* was used for mouse cells. PCR primer sequences are the following:

| Gene target | Primer sequence | Direction |
|---|---|---|
| hTP53 | CTCCTCTCCCCAGCCAAAGA | FW |
| hTP53 | GGAACATCTCGAAGCGCTCA | REV |
| hPHGDH | CTGCGGAAAGTGCTCATCAGT | FW |
| hPHGDH | TGGCAGAGCGAACAATAAGGC | REV |
| hPSAT1 | ACTTCCTGTCCAAGCCAGTGGA | FW |
| hPSAT1 | CTGCACCTTGTATTCCAGGACC | REV |
| hPSPH | GAGGACGCGGTGTCAGAAAT | FW |
| hPSPH | GGTTGCTCTGCTATGAGTCTCT | REV |
| hSLC7A5 | TCATCATCCGGCCTTCATCG | FW |
| hSLC7A5 | TCACGCTGTAGCAGTTCACG | REV |
| hSLC3A2 | CCAGGTTCGGGACATAGAGA | FW |
| hSLC3A2 | GAGTTAGTCCCCGAAATCAA | REV |
| hSLC1A5 | TGGTACGAAAATGTGGGCA | FW |
| hSLC1A5 | GTGCCCCAGCAGGCAGCACA | REV |
| hcMYC | AGGACGCGACTCTCCCGACG | FW |
| hcMYC | CGGCTGCACCGAGTCGTAGT | REV |
| mPSAT1 | CAGTGGAGCGCCAGAATAGAA | FW |
| mPSAT1 | CCTGTGCCCCTTCAAGGAG | REV |
| mPHGDH | TGGCCTCGGCAGAATTGGAAG | FW |
| mPHGDH | TGTCATTCAGCAAGCCTGTGGT | REV |
| mPSPH | GAGATGGAGCTACGGACATGGAAG | FW |
| mPSPH | CTCCTCCAGTTCTCCCAGCAGCTC | REV |
| mSLC7A5 | AGATGGGGAAGGACATGGGA | FW |

| mSL7A5 | GCCAACACAATGTTCCCCAC | REV |
|---|---|---|
| mSLC3A2 | GAAGATCAAGGTGGCGGAGGAC | FW |
| mSLC3A2 | CAAGTACTCCAGATGGCTCTTCAGACC | REV |
| mSLC1A5 | TGCTTTCGGGACCTCTTCTA | FW |
| mSLC1A5 | TGATGTGTTTGGCCACACCA | REV |

## Antibodies
The following antibodies and working concentrations were used for western blot: anti-HSP90 (1:10000, Santa Cruz Biotechnology sc-13119), anti-vinculin (1:10000, Sigma-Aldrich, v4505); anti-p53 DO-1 (1:1000 or 1:10000, Santa Cruz Biotechnology, sc-126); anti-LAT1/SLC7A5 (1:2000, Abcam, ab208776); anti-CD98hc/SLC3A2 (1:2000, Sigma-Aldrich, HPA017980); anti-ASCT2/SLC1A5 (1:2000, Cell Signaling Technology, #8057); anti-PSAT1 (1:2000, Proteintech, 10501-1-AP); anti-eIF2α D7D3 (1:5000, Cell Signaling Technology, #5324); anti-eIF2α phopsho S51 (1:1000, Cell Signaling Technology, #3398); anti-S6RP (1:1000, Cell Signaling Technology, #2317); anti-S6RP phospho S240/244 (1:1000, Cell Signaling Technology, #2215); anti-LC3 A/B (1:1000, Cell Signaling Technology, #12741); anti-HA (1:1000, Roche 1186742300); anti-Cleaved Caspase 3 (1:1000, Cell Signaling Technology, #9664).

The following antibodies and working concentrations were used for immunofluorescence analysis:

anti-p53 Valentino (1:100, homemade); anti-γH2AX S139 (1:100, Millipore 05-636); anti-4E-BP1 phospho T37/46 (1: 100, Cell Signaling Technology, #2855); anti-HA (1:100, Roche 1186742300).

The following primary antibodies and working concentrations were used for immunohistochemical stainings: anti-p53 (1:50 pH6, Leica Novocastra, NCL-L-p53-DO7), anti-LAT1/SLC7A5 (1:500 pH9, Abcam, ab208776), anti-PSAT1 (1:400 pH6, Proteintech, 10501-1-AP), anti-CD98hc/SLC3A2 (1:2500 pH9, Sigma Aldrich, HPA017980), anti-phospho-4E-BP1 (1:1000 pH9, Cell Signaling, #2855); anti-γH2AX S139 (1:300, Millipore, 05-636). Collagen fibers were stained with Picrosirius red stain (Bio Optica, 04-121873) as manufactuer's instructions.

## Protein extraction
For immunoblotting analyses total cell extracts were lysed in Lysis Buffer (NP40 1%, Tris-HCL pH=8 50 mM, NaCl 150 mM, EDTA 1 mM) solution, supplemented with PMFS 1 mM (Sigma-Aldrich), NaF 5 mM (Sigma-Aldrich), Na₃VO₄ 1 mM (Sigma-Aldrich), 10 μg/mL CLAP (Sigma-Aldrich). Protein concentration was determined with Bio-Rad Protein Assay Reagent (Bio-Rad). All the samples were denatured in Laemmli Sample Buffer 2x or 6X and finally by heating at 95 °C for 5 min.

## Western Blot analysis of mammalian cells
Proteins were loaded and separated in SDS-PAGE, followed by transferring on Nitrocellulose membranes (Cytiva). Blocking was performed in Blotto-tween (PBS, 0.2% Tween-20, non-fat dry milk 5%) or with TBST (0.2% Tween-20, Tris/HCl 25 mM pH 7.5) plus 5% non-fat dry milk or 5% BSA (PanReac Applichem) depending on the antibody. Anti-mouse and

anti-rabbit HRPO-conjugated (Sigma-Aldrich) were used as secondary antibodies. Membranes were analyzed by chemiluminescence using Pierce ECL™ Western Blotting Substrate or Pierce ECL™ Plus Western Blotting Substrate. Unprocessed scans of blots in figures and Supplementary figures are provided in the Source Data File.

## Immunofluorescence analysis of mammalian cells

Briefly, cells were fixed in 4% paraformaldehyde for 20 min, washed in phosphate-buffered saline (PBS), permeabilized with 0.1% Triton X-100 for 10 min, and blocked in 3% Fetal Bovine Serum FBS/PBS for 30 min. Antigen recognition was done by incubation with primary antibody at 4 °C for 14 h and with secondary antibodies (goat anti-mouse Alexa Fluor 568 and goat anti-rabbit Alexa Fluor 488, Life Technologies) at 4 °C for 1 h. Nuclei were stained with Hoechst (Life Technologies, 33342) or Dapi (Sigma Aldrich, 32670-F) for 15 min.

For immunofluorescence of MCF10DCIS.COM spheroids, they were fixed in 4% paraformaldehyde for 30 min, washed twice in PBS, permeabilized, and blocked in 0.5% Triton X-100/5% Horse Serum (HS)/PBS 14 h at 4 °C. The following day, antigen recognition was performed by incubation with primary antibody at 4 °C for 14 h on a rotating wheel and secondary antibodies (goat anti-mouse Alexa Fluor 568, goat anti-rabbit Alexa Fluor 488, goat anti-rat Alexa Fluor 647, Life Technologies) at room temperature for 2 h. Nuclei were stained with Dapi (Sigma Aldrich, 32670-F) for 15 min.

## Immunohistochemical analysis (IHC)

For IHC, human samples were fixed in 10% buffered formalin and paraffin-embedded. 4 mm-tissue sections were deparaffinized and rehydrated. Novocastra Epitope Retrieval Solution (pH6 or pH9) was used to unmask antigens in a thermostatic bath at 98 °C for 30 min. Subsequently, the sections were brought to room temperature and washed in PBS. After neutralizing the endogenous peroxidases with 3% $H_2O_2$ and Fcblocking by 0.4% casein in PBS (Novocastra), the sections were incubated with primary antibodies for 90 min at room temperature. The immunostaining was revealed by a polymer detection method (Novolink Polymer Detection Systems Novocastra Leica Biosystems Newcastle Ltd Product No: RE7280-K) and 3,3'-diaminobenzidine (DAB) substrate-chromogen. Slides were analyzed under a Zeiss Axioscope A1 microscope and microphotographs were collected using a Zeiss Axiocam 503 Color digital camera with the Zen 2.0 Software (Zeiss). Slide digitalization was performed using an Aperio CS2 digital scanner (Leica Biosystems) with the ImageScope software (Aperio ImageScope version 12.3.2.8013, Leica Biosystems). Quantitative analyses of IHC stainings were performed by calculating the average percentage of positive signals in five nonoverlapping fields at medium-power magnification (X200) using Nuclear Hub or Positive Pixel Count v9 ImageScope software.

## RNA-seq from MDA-MB-231 cell line

Two different RNA-seq experiments were performed in this work.

In first RNA-seq (Fig. 1b) MDA-MB-231 cells were transfected with siCTL or sip53 for 48 h. In the second RNA-seq (Fig. 5 and Suppl. Fig. 6) MDA-MB-231 cells were grown in complete medium (100% AAs) or in low AA medium (25% AAs) and transfected with siCTL or sip53 for 72 h. For total RNA extraction from the cells, the RNeasy Mini kit (Qiagen, 74104) was used with contaminant DNase treatment (RNase-Free DNase set, Qiagen, 79254).

mRNA libraries were obtained by the Illumina TruSeq stranded mRNA library construction kit. mRNA libraries were sequenced using Illumina Novaseq 6000 for 100 bp paired-end sequencing (60 M reads).

## Analysis of RNA-seq data from MDA-MB-231 cell line

Read quality was verified using fast QC (version 0.11.3; http://www.bioinformatics.babraham.ac.uk/projects/fastqc/). Raw reads were trimmed for adapters, polyA read-through, and low-quality tails (quality <Q20) using BBDuk (version 37.02; sourceforge.net/projects/bbmap/). Reads shorter than 35 bp were also removed resulting on average 51 M trimmed reads per sample. Reads were subsequently aligned to the human reference genome (hg38) using STAR (version 2.7.3a[91]); with default parameters. Raw gene counts were obtained using the *featureCounts* function of the *Rsubread* R package (version 2.0.1[92]); and the Gencode gene annotation for hg38 genome. Gene counts of the first RNA-seq (Fig. 1b) were normalized to counts per million mapped reads (cpm) using the *edgeR* package (version 3.28.1[93]); only genes with a CPM greater than 1 in at least 2 samples were further retained for differential analysis Gene counts of second RNA-seq (Fig. 5 and Suppl. Fig. 6) were normalized to z-scores subtracting the mean expression across samples and dividing it for the standard deviation of each single gene. Differential gene expression analysis was performed using the *DESeq* function of the *DESeq2* package (version 1.36[94]);. Over-representation analysis was performed using Gene Set Enrichment Analysis and gene sets of the KEGG and Reactome collections from the Broad Institute Molecular Signatures Database (http://software.broadinstitute.org/gsea/msigdb). GSEA software (http://www.broadinstitute.org/gsea/index.jsp) was applied on expression data of cells transfected with mutant-p53 or with a control vector. Gene sets were considered significantly enriched at FDR < 5% when using Signal2Noise as a metric and 1,000 permutations of gene sets. Gene expression heatmap of Fig. 1b has been generated calculating the log2 of cpm expression values and after row-wise standardization. Gene expression heatmap of Suppl. Fig. 6i has been generated using z-scores expression values. All analyses were performed using R 4.2.1 and publicly available packages explicitly cited in the manuscript. No custom functions were written for the analysis.

## Gene expression datasets

Gene expression data (raw counts), TP53 mutations, and clinical information of the TCGA breast cancer dataset (TCGA-BRCA) were downloaded from the Genomic Data Commons Portal using functions of the *TCGAbiolinks* R package (version 2.23.1[95]);. Raw counts were normalized, and gene expression levels were quantified as count per million (cpm), fragments per kilobase of exon per million mapped reads (fpkm) using functions of the *edgeR* R package (version 3.32.1[93]); and z-score subtracting the mean expression across samples and dividing it for the standard deviation of each single gene. The METABRIC collection, comprising gene expression data and clinical annotations for 997 breast cancer samples, has been downloaded from the European Genome-Phenome Archive (EGA, http://www.ebi.ac.uk/ega/) under accession number EGAD00010000210[96]. Original Illumina probe identifiers have been mapped to Entrez gene IDs using the Bioconductor illuminaHumanv3.db annotation package for Illumina HT-12 v3 arrays obtaining log2 intensity values for a total of 19,761 genes. The TP53 status of 701 samples annotated as wild-type ($n = 584$) and 'missense' mutant p53 ($n = 117$) was derived from Silwal-Pandit et al.[97].

## Tumor classification based on signature scores

Genes and gene sets expression levels have been calculated as the average expression of single genes or all gene set genes in sample subgroups (i.e., TP53 status). Genes up-regulated by the activation of mTORC1 complex have been downloaded from the Molecular Signature Database (MSigDB) mTORC1 gene set (https://www.gsea-msigdb.org/gsea/msigdb/cards/HALLMARK_MTORC1_SIGNALING).

The "mutp53 AA metabolism" signature has been derived from the intersection of significant upregulated DEGs calculated from the comparison "siCTL Low AA - siCTL complete medium" and down-regulated DEGs calculated from the comparison "sip53 Low AA- siCTL Low AA" (Suppl. Table 6).

## Single cell RNA sequencing (scRNAseq) library preparation from mutp53 and wtp53 mice

Mammary gland from 8-week-old mice (one p53[R172H/R172H] knock-in mouse and one p53[+/+] mouse) were isolated and digested as previously described[98]. scRNAseq libraries were prepared by using Chromium Next GEM Single Cell 3' Reagent Kits v3.1 Dual Index (10X Genomics) following manufacturer's instructions. Estimated 16,500 cells were loaded to each channel with the average recovery rate of 10,000 cells. Briefly, cells were resuspended in the master mix and loaded together with partitioning oil and gel beads into the chip to generate the gel bead-in-emulsion (GEM). The poly-A RNA from the cell lysate contained in every single GEM was retrotranscribed to cDNA, which contains an Illumina R1 primer sequence, Unique Molecular Identifier (UMI) and the 10x Barcode. The pooled barcoded cDNA was then cleaned up with Silane DynaBeads, amplified by PCR and the appropriate size fragments were selected with SPRIselect reagent for subsequent library construction. During the library construction Illumina R2 primer sequence, paired-end constructs with P5 and P7 sequences and a sample index were added. The final indexed libraries were quantified by using the Cell-free DNA Screen Tape Assay (Agilent) on a TapeStation 4150 (Agilent) and were sequenced on HiSeqX (Illumina).

## Processing and quality control of scRNAseq data from mutp53 and wtp53 mice

FASTQ files were aligned utilizing 10x Genomics Cell Ranger v.6.1.1. Each library was aligned to an indexed mm39 genome using Cell Ranger Count. We applied stringent filters to eliminate cells with UMI counts <1250 or >37000, gene counts <700, and mitochondrial gene ratio>10% and with UMI counts <1750 or >50,000, gene counts <800, and mitochondrial gene ratio >10% for the wtp53 and mutp53 respectively. This pre-filtering resulted in the detection of 3206 and 3442 cells for the wtp53 and mutp53 respectively.

Cell doublets were estimated using *Scrublet*[99]. For each individual sample, we ran *Scrublet* with default parameters using the raw count data from CellRanger output. We then visualized the predicted singlets and doublets on the UMAP space. One small group of cells was predicted as potential doublets; we did remove them from subsequent analysis and remain with a total of 3169 and 3382 cells for the wtp53 and mutp53 respectively. Cell cycle estimation was performed in Seurat using the default cell-cycle gene signatures.

## Integration and clustering analysis of scRNAseq data from mutp53 and wtp53 mice

Clustering of cells was performed using the Seurat R package[100] on the integrated data. Briefly, single-cell data matrices were column-normalized and log-transformed. To identify cell clusters, principal component analysis (PCA) was first performed and the top 20 PCs with a resolution of 0.6 were used to obtain 21 clusters. To present high-dimensional data in two-dimensional space, we performed UMAP analysis using the results of PCA with the top 20 significant PCs as input. After data integration, cell type identification was performed using the expression of known marker genes (Suppl. Table 2). For subclustering of epithelial cells, we subset the epithelial clusters from mutp53 or wtp53 replicate using the R *subset* internal function.

Differential gene expression analysis was performed to compare epithelial cells from mutp53 and wtp53 samples by using Seurat's *FindMarkers* function (Wilcoxon rank sum test).

Differentially expressed genes were identified having an expression difference of at least 1-logfold and a *p-value* of <0.05. Pathway analysis was performed using the GSEA software on all unfiltered differentially expressed genes using MSigDB v7.0[101,102] curated canonical pathway database, specifically Gene Ontology (GO) and Reactome. To ensure proper gene name mapping, all mouse gene names were converted to their human homologs using NCBI Homologene prior to

analysis in GSEA. All the top enriched pathways regarding glucose, amino acids and lipids metabolism were shown.

## Analysis of ChIP-seq data

To retrieve mutp53 binding sites, we used ReMap2022[43], we chose MDA-MB-231 cell experiments as a cell model, based on GEO GSE95303 data, and in particular the experiment GSM2501568. Moreover, we have analyzed ChIP-seq data regarding histone modifications H3K4me3 and H3K27Ac from the GEO GSE49651 regarding the same cell line.

Consequently, we have searched for TP53 peaks across the genome selecting genomic regions of interest and therefore retrieving mutp53 peaks in our genes of interest (Suppl. Table 1). After that, we checked if there were histone modifications indicating open chromatin in the same regions.

Moreover, by using IGV (Integrative Genomics Viewer, version 2.11.3)[103], we have inspected ChIP-seq data from ReMap2022 looking for TFs that bound the same regions of *TP53* peaks on *PSAT1*, *SLC7A5* and *SLC3A2* promoters in the MDA-MB-231 cell-line, considering a region extended by ±1000 nucleotides upstream or downstream the selected. Among these, we selected TFs that bind to all these genes in the same genomic loci of mutp53.

## Chromatin Immunoprecipitation (ChIP)

Chromatin immunoprecipitation was performed as previously described[61,62]. Briefly, cells were lysed in lysis buffer−50 mM HEPES pH 7.9, 140 mM NaCl, 1 mM EDTA, 10% glycerol, 0.5% NP-0.4, 0.25% Triton X-100, nuclei spun down, washed in 10 mM Tris-HCl, pH 7.5, 200 mM NaCl, 1 mM EDTA and resuspended in shearing buffer−0.1% SDS, 1 mM EDTA, 10 mM Tris, pH 7.5. Samples were sonicated using a Bioruptor sonicator (Diagenode; medium power setting) for a total time of 30 min, to achieve an average size of 250−300 bp of the sonicated chromatin fragments. The shearing buffer was then supplemented to obtain the RIPA buffer of the composition as described in Girardini et al., 2011[62] and the rest of the protocol was followed. Chromatin was immunoprecipitated with the p53 DO-1 (sc-126, Santa Cruz Biotechnology) antibody. Co-immunoprecipitated DNA was analyzed by real-time PCR. Promoter occupancy was calculated as the percentage of input chromatin immunoprecipitated using the $2 - \Delta Ct$ method. Primer sequences are reported in the following table.

| Gene target | Primer sequence | Direction |
|---|---|---|
| hPSAT1 | GCGAACCAATTAGCGCAGGG | FW |
| hPSAT1 | TCAGCCAAGGAGGACCGAA | RV |
| hSLC7A5 | AGGGCTTGTCATTCTGGACC | FW |
| hSLC7A5 | TATCTGTGTGACCTCCGCAC | RV |
| hSLC3A2 | ATGGAGCTACAGCCTCCTGA | FW |
| hSLC3A2 | TACTTTCCCGAAATGGCCCC | RV |

## Metabolite extraction and liquid chromatography/tandem mass spectrometry analysis

For metabolic tracing analyses, cells were exposed for 24 h to 12.5 mM [U-13C6] glucose (Sigma-Aldrich, 389374) or 30 min to 802 μM [U-13C6] Leucine (Sigma-Aldrich, 605239).

Metabolite extraction and analysis were performed as previously described[104]. Briefly, cells were harvested in 250 μl of ice-cold methanol/acetonitrile 1:1 and spun at 20,000 × g for 5 min at 4 °C. Supernatants were then passed through a regenerated cellulose filter, dried, and resuspended in 100 μl methanol for subsequent analysis. Methanolic samples were analyzed by 10 min runs in positive (amino acids) and 5 min runs in negative (energy metabolites) ion mode with 35-multiple-reaction monitoring (MRM) transition in positive ion mode and a 139-MRM transition in negative ion mode to analyze the different isotopomers, respectively.

For targeted metabolomic analysis, cells were extracted using tissue lyser for 30 s at maximum speed in 250 μL of ice-cold methanol: water: acetonitrile 55:25:20 containing [U-$^{13}C_6$]-glucose 1 ng/μL and [U-$^{13}C_5$]-glutamine 1 ng/μL as internal standards (Merk Life Science, Milan, Italy). Lysates were spun at 15,000 g for 15 min at 4 °C, dried under $N_2$ flow at 40 °C and resuspended in 125 μL of ice-cold methanol/ water 70:30 for subsequent analyses.

Amino acids analysis was performed through the previous derivatization. Briefly, 50 μl of 5% phenyl isothiocyanate in 31.5% ethanol and 31.5% pyridine in water were added to 10 μl of each sample. Mixtures were then incubated with phenyl isothiocyanate solution for 20 min at room temperature, dried under $N_2$ flow, and suspended in 100 μl of 5 mM ammonium acetate in methanol/ H2O 1:1. Quantification of different amino acids was performed by using a C18 column (Biocrates, Innsbruck, Austria) maintained at 50 °C. The mobile phases were phase A: 0.2% formic acid in water and phase B: 0.2% formic acid in acetonitrile. The gradient was $T_0$: 100% A, $T_{5.5}$: 5% A and $T_7$: 100% A with a flow rate of 500 μL/min.

Measurement of energy metabolites and cofactors was performed by using a cyano-phase LUNA column (50 mm × 4.6 mm, 5 μm; Phenomenex, Bologna, Italy), maintained at 53 °C, by a 5 min run in negative ion mode. The mobile phase A was water, while phase B was 2 mM ammonium acetate in MeOH, and the gradient was 50% A and 50% B for the whole analysis, with a flow rate of 500 μL/min.

Acylcarnitines quantification was performed using a ZORBAX SB-CN 2.1x150mm, 5 μm column (Agilent, Milan, Italy). Samples were analyzed by a 10 min run in positive ion mode. The mobile phases were phase A: 0.2% formic acid in water and phase B: 0.2% formic acid in acetonitrile. The gradient was The gradient was $T_0$: 100% A, $T_{5.5}$: 5% A and $T_6$: 100% A with a flow rate of 350 μL/min.

All metabolites analyzed were previously validated by pure standards, and internal standards (only in targeted metabolomics analysis) were used to check instrument sensitivity.

All data were acquired on an API-4000 triple quadrupole mass spectrometer (AB Sciex) coupled with an HPLC system (Agilent Technologies), a CTC PAL HTS autosampler (PAL System), and Triple Quad 3500 (AB Sciex) with an HPLC system (AB Sciex) and a built-in autosampler (AB Sciex). MultiQuant software (version 3.0.2) was used for data analysis and peak review of chromatograms. Data processing, normalization and analysis were performed using MetaboAnalyst 5.0 web tool as previously described[105].

## Proliferation assay

Cells ($1,2 \times 10^4$ to $4 \times 10^4$ cells/well depending on the cell lines) were plated in 12-well plates in their regular medium. The next day, after washing cells with PBS, the medium was changed with medium with different amino acids concentration as indicated in the figures. At 3 or 6 days following the treatment, cells were trypsinized, suspended in medium and counted. The relative cell number was calculated based on the number of cells initially plated. Proliferation rate was determined using the following formula: Proliferation rate (doublings/day) = [$\log_2$(Final Day 3 cell number/Initial Day 0 cell number)]/3 days[12]. For each experimental condition, two technical replicates were plated and counted.

## BrdU incorporation assay

4T1 cells ($1,2 \times 10^4$) were plated in 12-well plates in presence of doxycycline. After 48 h, the DNA precursor bromodeoxyuridine (BrdU) (20 μM) was added to the medium for 3 h before fixation.

For the experiment in low AA, the day after plating cells were washed with PBS and the medium was changed with medium with reduced amino acids concentration. At 36 h following the treatment, BrdU (20 μM) was added to the medium for 12 h before fixation. Briefly, the cells were fixed in 4% paraformaldehyde for 20 min, washed in PBS, permeabilized with Triton 0.1% for 10 min, and washed three times with

NaOH 50 mM solution and washed in PBS. Primary anti-BrdU antibody solution (1:2 dilution), to detect incorporated BrdU, was used for 2 h at 37 °C and goat anti-mouse Alexa Fluor 568 (Life Technologies) was used as the secondary antibody for 1 h a 37 °C. Nuclei were counterstained with Hoechst 33342 (Life Technologies). The percentage of cells incorporating BrdU out of 100–150 nuclei/condition in immunostained samples was determined by fluorescence microscopy.

For BrdU assay in DCIS spheroids, BrdU (20 μM) was added to the medium for 12 h before fixation. Briefly, the cells were fixed in 4% paraformaldehyde for 20 min, washed in PBS, permeabilized with Triton 0.5%/5% HS/PBS for 14 h at 4 °C. and washed three times with NaOH 50 mM solution, and washed in PBS. Primary anti-BrdU antibody solution (1:2 dilution), to detect incorporated BrdU, was used for 14 h at 4 °C and goat anti-mouse Alexa Fluor 568 (Life Technologies) was used as the secondary antibody for 2 h at room temperature. Nuclei were stained with Dapi (Sigma Aldrich, 32670-F) for 15 min. The percentage of cells incorporating BrdU out of 900 nuclei/ spheroid in immunostained samples was determined by fluorescence microscopy.

## Colony formation assay

Cells ($5 \times 10^2$ to $4 \times 10^3$ cells/well depending on the cell lines) were plated in 6/12-well plates. The next day, after washing cells with PBS, the medium was changed with medium with 25% of amino acids and upon treatments as indicated in the figures. Growth continued until the appearance of clones (at least 1 week). Briefly, the cells were fixed in 4% paraformaldehyde for 20 min, washed in PBS, and stained with GIEMSA solution (1:10 in water) for 1 h. Colonies ≥1000 pixels were counted using countPHICS software after background subtraction.

## In vivo procedures

6- to 8-week-old female BALB/c mice were injected with: $1 \times 10^5$ or $5 \times 10^4$ 4T1 TetOn clone cells in PBS orthotopically into the mammary fat pad, under general anesthesia. Doxycycline was given in their drinking water to induce mutp53 expression. After palpable tumors were detected, mice were treated with NCT503 (38 mg/kg) and JPH203 (20 mg/kg). Tumor growth was monitored every 2 to 3 days by caliper measurements and the volume was calculated using the formula: tumor volume (mm3) =Dxd2/2, where D and d are the longest and the shortest diameters, respectively. At 24 days after the injection, mice were euthanized, and primary tumors were extracted and directly frozen in liquid nitrogen for molecular analyses.

All mice were housed in ventilated cages, monitored daily by ICGEB BioExperimentation Facility staff, and had food and water ad libitum. Mice were weighed at least two times per week.

Procedures involving animals and their care were in conformity with national (D. L. 26/2014 and subsequent implementing circulars) and international (EU Directive 2010/63/EU for animal experiments) laws and policies, and the experimental protocol (Authorization n. 347/2022-PR) was approved by the Ethical Committee of the ICGEB and by the Italian Ministry of Health.

## Statistics

All the experiments are representative of at least three independent repeats. Graph bars represent mean±s.d. from at least $n = 3$ biological replicates. $P$ values were determined using the following statistic tests as indicated in figure legends: two-tailed Student's $t$-test, with a 95% confidence threshold; One-way ANOVA (Fisher's LSD) ($\alpha < 0.05$); Exact test $p$-value (FDR) ($\alpha < 0.05$); Wilcoxon Rank Sum test ($p$val) and Bonferroni correction ($p$adj) ($\alpha < 0.05$); Wald test and Bonferroni correction ($p$adj) ($\alpha < 0.05$).

## Reporting summary

Further information on research design is available in the Nature Portfolio Reporting Summary linked to this article.

## Data availability

Data generated in this study are available within this paper and upon request from the Lead Contact. The RNAseq datasets generated during this study have been deposited to GEO, accession number GSE214494. scRNAseq datasets generated during this study have been deposited to GEO, accession number GSE239706. Any additional information required to reanalyze the data reported in this work paper is available from the Corresponding author upon request. Source data are provided with this paper.

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

## Acknowledgements

We thank M. Foiani (IFOM institute, Milan), C. Vernieri (IFOM institute, Milan), K.Havas (IFOM institute, Milan) for scientific discussions and suggestions and A. Testa for reading and editing the manuscript. We acknowledge G. Pastore for technical support and we thank the Bioexperimentation Unit of the ICGEB. We acknowledge support by the Italian University and Research Ministry PRIN-2017HWTP2K_004 and MIUR-ARSO1_00876, the Italian Ministry of Health RF-2019-12368718, and grants from the European Union, European Regional Development Fund and Interreg V-A Italia-Austria 2014–2020 (project code ITAT1050), the Fondazione AIRC IG grant 22174 to G.D.S. and the Fondazione AIRC Special Program Molecular Clinical Oncology "5 per mille" grant 22759 to G.D.S, G.BI., S.B. and C.T. This work was partially supported by the Ministry of University and Research (MUR) Progetto Eccellenza (2023–2027) to the Dipartimento di Scienze Farmacologiche e Biomolecolari, Università degli Studi di Milano to M.A., S.P. and N.M. and partially by the Italian Ministry of Health with Ricerca Corrente and "5xmille" funds to N.M., B.B. is supported by the Fondazione AIRC IG grant 20061. G.BA. is supported by Italian Health Ministry Ricerca Corrente (linea 1). We are very grateful to the patients who consented to donating their samples.

## Author contributions

C.T., A.Z. and G.D.S. conceived the study. A.Z. and C.T. designed experiments. A.Z., C.T., R.B. and M.C. performed the experiments and analyzed the data. F.M. supervised the ChIP experiments and the generation of inducible transduced cell lines. M.A., S.P. and N.M. performed metabolomic and metabolic tracing analyses. L.T., S.P. and S.B. performed bioinformatic analyses. V.C., D.V. and C.T. performed immunohistochemistry on mouse and human tumor sections. S.V., A.Z., C.T., R.B. and A.R. performed mice experiments and analyzed the data. Breast Cancer specimens for PDOs generation were provided by S.D., G.Bl., I.S., B.B. and G.BA., PDOs have been generated by I.S., A.Z. and C.T. A.Z., C.T., R.B., F.M. and G.D.S. wrote the manuscript.

## Competing interests

The authors declare no competing interests.
