## [Peer review file · Nature Communications]

REVIEWERS' COMMENTS

Reviewer #1 (Remarks to the Author):

The manuscript by the Del Sal group describes a very novel mechanism for mutant p53 in breast cancer growth. The data are clean, beautiful, and very convincing. There is one figure that needs some correction.

Figure 6c - the tumors on the right need a ruler, since they are part of a different photo. The tumors on the left - in the top row, the last tumor is not part of the same photo. It would be nice to have them as part of the same photo or another ruler is needed to indicate the size.

Reviewer #2 (Remarks to the Author):

Tombari, C. et al. investigated metabolic impact of mutp53 that was not thoroughly studied. The authors found that the upregulation of serine-glycine-one-carbon metabolism pathway and intake of essential AAs enable breast cancer cells with mutp53 better survive the nutrient limited environment. Overall, the interaction of mutp53-SGOC has been established and such finding is valuable considering the importance of p53. The manuscript has been improved compared to the initial submission.

Two points need to be addressed. 1. there are many "amino acid deprivation" statement throughout the manuscript but in most scenario if not all. it should be a "restriction" or "limitation" instead of deprivation since the conditions used are "lower AA".

2. there is no fig 1h.

Reviewer #3 (Remarks to the Author):

The authors addressed almost all of my comments and significantly strengthened the manuscript. However, I would ask a couple of things to be further clarified:

1. Are experiments presented in Figure 1a and ED figure 1b the same set of experiments? 1a shows that sip53 has less glucose-derived serine and glycine synthesis and it is statistically significant. Is this the case in ED fig 1b? Is this difference (sip53 vs siCTL) also statistically significant in Figure 4i.

What kind of values MID (mass isotopomer distribution) are? Is it fractional enrichment or normalised abundance?

2. The authors now also performed the experiment demonstrating the increase of serine and glycine production from glucose upon serine and glycine deprivation and that this increase is inhibited by p53 ND (ED Fig 1b). What happens with cell proliferation in this experiment (this is related to my comment about serine and glycine deprivation only in the initial review)?

Reviewer #1

The manuscript by the Del Sal group describes a very novel mechanism for mutant p53 in breast cancer growth. The data are clean, beautiful, and very convincing. There is one figure that needs some correction. *We thank the reviewer for his/her positive comments about our work.*

Figure 6c - the tumors on the right need a ruler, since they are part of a different photo. The tumors on the left - in the top row, the last tumor is not part of the same photo. It would be nice to have them as part of the same photo or another ruler is needed to indicate the size.

The images provided are representative of the in vivo experiment and quantified in Fig. 6d. We now have modified the figure according to the reviewer request, showing the ruler for each picture.

Reviewer #2

Tombari, C. et al. investigated metabolic impact of mutp53 that was not thoroughly studied. The authors found that the upregulation of serine-glycine-one-carbon metabolism pathway and intake of essential AAs enable breast cancer cells with mutp53 better survive the nutrient limited environment. Overall, the interaction of mutp53-SGOC has been established and such finding is valuable considering the importance of p53. The manuscript has been improved compared to the initial submission.

We thank the reviewer for his/her positive evaluation of our study.

Two points need to be addressed. 1. there are many "amino acid deprivation" statement throughout the manuscript but in most scenario if not all. it should be a "restriction" or "limitation" instead of deprivation since the conditions used are "lower AA".

We agree with the reviewer, and we have modified the text accordingly.

2. there is no fig 1h.

We have now corrected the mistake.

Reviewer #3

The authors addressed almost all of my comments and significantly strengthened the manuscript. However, I would ask a couple of things to be further clarified:

We thank the reviewer for his/her suggestions, and we have now improved the MS.

1. Are experiments presented in Figure 1a and ED figure 1b the same set of experiments?

1a shows that sip53 has less glucose-derived serine and glycine synthesis and it is statistically significant. Is this the case in ED fig 1b? Is this difference (sip53 vs siCTL) also statistically significant in Figure 4i.

The experiments in Fig.1a and ED Fig. 1b are two independent set of experiments performed under the same conditions, as described in the text. In the experiment shown in ED Fig. 1b, although we observed a decrease in glucose-derived serine and glycine, this is not statistically significant, likely due to lower number of replicates.

In Fig 4i, the difference observed between sip53 vs siCTL in CM is not statistically significant. In these experiments mutp53 was already silenced 24 hours before treatment with low AA for 72 hours. Hence, cells were silenced for a total of 96 hours rather than 48, as in Fig. 1a. It is possible that this prolonged period of silencing could have masked the acute metabolic effect seen instead after 48 hours.

What kind of values MID (mass isotopomer distribution) are? Is it fractional enrichment or normalised abundance?

A mass isotopomer distribution (MID) quantitates the relative abundance of mass isotopomers of a metabolite. The fractional abundance (FA) of the “I”-mass isotopomer in a MID (FAM_i), is calculated as follow:

$$FAM_i = \frac{Im_i}{\sum_{k=0}^n Im_k}$$

where *i* is the incremental increase in atomic mass, *Im_i* is the measured spectral intensity obtained at a specific mass-to-charge ratio corresponding to an *M₀+i* mass shift, and *n* is the total number of possible mass isotopomers for a given metabolite. The sum of all FAM_i values in a MID is 1.

On this ground, MID value of each isotopomer represents its fractional enrichment on a total of 1 (Antoniewicz, M.R. A guide to ¹³C metabolic flux analysis for the cancer biologist. *Exp Mol Med* 50, 1–13 (2018). <https://doi.org/10.1038/s12276-018-0060-y>).

2. The authors now also performed the experiment demonstrating the increase of serine and glycine production from glucose upon serine and glycine deprivation and that this increase is inhibited by p53 ND (ED Fig 1b). What happens with cell proliferation in this experiment (this is related to my comment about serine and glycine deprivation only in the initial review)?

We thank the reviewer for the question. Unfortunately we did not measure the proliferation rate of cells under these conditions, since this experiment (ED Fig 1b) was only a control to evaluate the metabolic effects of mutp53 in a condition where SSP was strongly stimulated. Nevertheless, we noticed a clear reduction of proliferation under sip53 -S/G condition, suggesting that ablation of mutp53 in BC cells increased the sensitivity to S/G deprivation.